# EXPLORE-EXECUTE CHAIN: TOWARDS AN EFFICIENT STRUCTURED REASONING PARADIGM

## ABSTRACT

Chain-of-Thought (CoT) and its variants have markedly advanced the reasoning abilities of Large Language Models (LLMs), yet their monolithic and auto-regressive architecture inherently conflates high-level strategic planning with low-level step-by-step execution, leading to computational inefficiency, limited exploration of reasoning paths, and reduced interpretability. To overcome these issues, we propose the **Explore-Execute Chain ($E^2C$)**, a structured reasoning framework that decouples reasoning into two distinct phases: an exploratory phase that stochastically generates succinct high-level plans, followed by an execution phase that deterministically carries out the chosen plan. Our approach incorporates a two-stage training methodology, which combines Supervised Fine-Tuning (SFT)—augmented by a novel data generation algorithm enforcing strict plan adherence—with a subsequent Reinforcement Learning (RL) stage that capitalizes on the informativeness of exploration and reinforces the determinism of execution.This decomposition enables an efficient test-time scaling strategy: on AIME'2024, **$E^2C$ Test Time Scaling** reaches 58.1% accuracy using <10% of the decoding tokens required by comparable methods (e.g., Forest-of-Thought), sharply cutting self-consistency overhead. For cross-domain adaptation, our **Exploration-Focused SFT (EF-SFT)** fine-tunes with only 3.5% of the tokens used by standard SFT yet yields up to 14.5% higher accuracy than **standard SFT** on medical benchmarks, delivering state-of-the-art performance, strong generalization, and greater interpretability by separating planning from execution.

## 1 INTRODUCTION

Large Language Models (LLMs) have demonstrated remarkable capabilities in complex reasoning, largely propelled by techniques such as Chain-of-Thought (CoT) prompting (Wei et al., 2022). This paradigm has inspired a suite of advanced methods, including sampling multiple reasoning paths for consensus via Self-Consistency (Wang et al., 2022), and exploring the solution space with more complex structures like Tree-of-Thoughts (ToT) (Yao et al., 2023), Graph-of-Thoughts (GoT) (Besta et al., 2023), and Forest-of-Thought (FoT) (Bi et al., 2025). Other approaches focus on iterative refinement through self-correction (Shinn et al., 2023) or problem decomposition (Zhou et al., 2023; Yao et al., 2022).

Despite their success, these methods are predominantly founded on a monolithic, auto-regressive generation process that conflates two fundamentally different cognitive functions: high-level strategic **planning** and low-level, step-by-step **execution**. This entanglement leads to critical inefficiencies. First, the model expends equivalent computational effort on both creative planning and routine calculations, a challenge addressed by works on adaptive computation (Xu et al., 2025b) and reasoning compression (Li et al., 2025). Second, the greedy generation process restricts the diversity of initial strategies, where a suboptimal early choice can derail the entire reasoning path. This is a key problem that sophisticated test-time scaling methods (Liao et al., 2025; Xu et al., 2025a) and structured exploration frameworks (Zheng et al., 2025a) aim to mitigate.

In this work, we argue that explicitly decoupling these two functions is crucial for advancing reasoning in large language models. We introduce the **Explore–Execute Chain ($E^2C$)**, a framework that decomposes standard CoT into two distinct phases. The first phase is a highly informative **exploration** stage, in which the model generates a concise, high-level plan. This stage provides a quick preview of the complete reasoning process—analogous to hierarchical planning (Gui et al.,

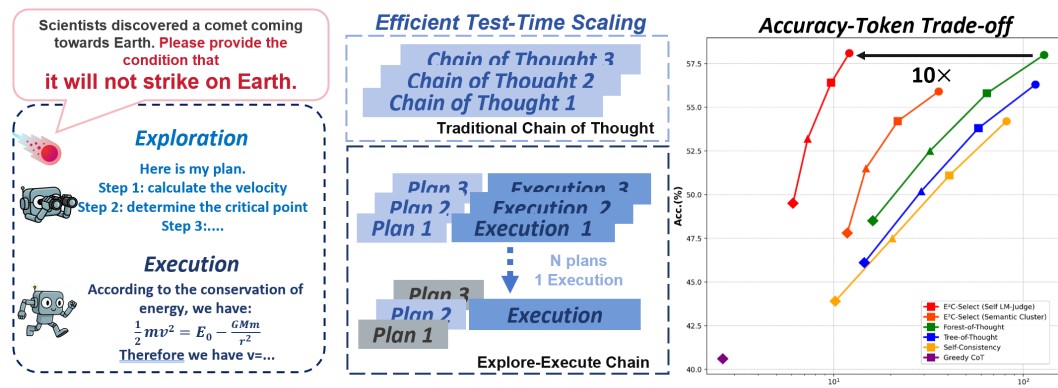

Figure 1: Our proposed **Explore-Execute Chain (E²C)** method decomposes reasoning chains into a short, high-level exploratory plan followed by a long, detailed execution (left). After optimizing these special reasoning chains using RL, it is possible to synthesize a large number of plans, use the model to pick the best plan, and then execute this plan (middle). This unlocks dramatically improved overall token efficiency on the challenging AIME'2024 benchmark (right).

2025)—without incurring the cost of full-chain generation. The second phase is a highly deterministic **execution** stage, which takes the plan as guidance and meticulously performs the detailed calculations. This stage emphasizes precision and faithful adherence to the chosen strategy, a requirement that necessitates specialized training (Zheng et al., 2025b).

This decomposition enables a highly efficient **test-time scaling** strategy (Fig. 1). Rather than generating multiple costly, full reasoning chains (Wang et al., 2022), E²C samples a larger set of inexpensive **exploration** plans while executing fewer **execution** steps. The most promising exploration plans are selected via semantic clustering or an LLM, leveraging the high informativeness of the exploration phase for effective filtering. The chosen plan is then executed with high determinism, ensuring reliable and precise reasoning. This approach improves the performance–cost trade-off (Geiping et al., 2025; Liao et al., 2025) and enhances interpretability. We implement this framework using a two-stage (SFT+RL) training pipeline, guided by recent advances in reasoning alignment (Gan et al., 2025; Rafailov et al., 2023).

Our main contributions are summarized as follows:

- We propose the Explore–Execute Chain (E²C), which decouples LLMs' reasoning into a highly informative Exploration stage for planning and a highly deterministic Execution stage for carrying out the plan, thereby improving efficiency and interpretability.

- We introduce a robust two-stage training methodology (SFT+RL) together with a specialized data construction algorithm that ensures the model faithfully adheres to its plans, effectively instilling E²C paradigm and achieving superior performance.

- We demonstrate the efficiency of this framework with two key results: an efficient test-time scaling strategy that achieves 58.1% accuracy on AIME'2024 using less than 10% of the decoding tokens required by comparable methods (e.g., Forest-of-Thought); and a data-efficient, robust domain-adaptation method—Exploration-Focused SFT (EF-SFT)-that, with only 3.5% of the tokens used by standard SFT, improves medical benchmark performance by up to 14.5% over standard SFT.

## 2 RELATED WORK

In this work, we argue that explicitly decoupling these two functions is crucial for advancing reasoning in large language models. We introduce the **Explore–Execute Chain (E²C)**, a framework that decomposes standard CoT into two distinct phases. The first phase is a highly informative **exploration** stage, in which the model generates a concise, high-level plan. This stage provides a quick preview of the complete reasoning process—analogous to hierarchical planning (Gui et al., 2025)—without incurring the cost of full-chain generation. The second phase is a highly deterministic **execution** stage, which takes the plan as guidance and meticulously performs the detailed

calculations. This stage emphasizes precision and faithful adherence to the chosen strategy, a requirement that necessitates specialized training (Zheng et al., 2025b).

**From Chain-of-Thought to Structured Reasoning:** Chain-of-Thought (CoT) prompting (Wei et al., 2022) significantly improves LLM reasoning, but its linear nature has motivated more robust structured paradigms that explore diverse reasoning paths (Chen et al., 2025). These include parallel sampling methods such as Self-Consistency (Wang et al., 2022; Wan et al., 2025), and more complex search structures including trees (ToT) (Yao et al., 2023), graphs (GoT) (Besta et al., 2023; Yao et al., 2024), and forests (FoT) (Bi et al., 2025). Further advances involve RL-trained parallel thinking (Zheng et al., 2025b; Pan et al., 2025; Yang et al., 2025b) and hierarchical decomposition via hypertrees (Gui et al., 2025). While these paradigms expand the search space—often integrating algorithms like MCTS (Zhang et al., 2024; Xie et al., 2024)—they often conflate high-level planning with low-level execution. $E^2C$ addresses this limitation through explicit decoupling.

**Planning and Decomposition in LLM Reasoning:** The core idea of separating planning from execution in $E^2C$ aligns with a growing body of work on task decomposition. Methods range from breaking problems into subtasks (Zhou et al., 2023; Press et al., 2022) to interleaving reasoning with tool use (Yao et al., 2022; Schick et al., 2023; Patil et al., 2023). Hu et al. (2025) leveraged learned belief states to improve planning. Wang et al. (2024a) introduce *PlanSearch* to enhance performance in code generation tasks. While many approaches rely on LLMs as planners for external solvers (Hao et al., 2023; Liu et al., 2023) or within multi-agent systems (Yuan et al., 2024), $E^2C$ inherently supports explore–execute reasoning, yielding greater stability during inference. Moreover, by exploiting this decomposition property in training, $E^2C$ achieves superior performance.

**Test-Time Scaling and Reasoning Efficiency:** Test-time scaling (TTS) aims to improve performance by increasing inference-time compute (Snell et al., 2024; Wu et al., 2025), but methods like Self-Consistency (Wang et al., 2022) are costly because they generate multiple full-length solutions. This has spurred research on reasoning efficiency, including CoT compression via step entropy (Li et al., 2025) or truncation (Liao et al., 2025), and adaptive termination guided by semantic entropy to avoid redundant computation (Xu et al., 2025b). Other efficiency-driven directions include entropy-guided RL exploration (Zheng et al., 2025a) and reasoning in a continuous latent space (Geiping et al., 2025; Xu et al., 2025a; Hao et al., 2024). Training these capabilities via Reinforcement Learning from Verifiable Rewards (RLVR) has also become a key area (Guo et al., 2025; Yue et al., 2025; Yu et al., 2025; Shao et al., 2025). $E^2C$ contributes a novel TTS strategy: it samples multiple inexpensive plans and executes only the most promising one, thereby achieving ensembling-like gains at a fraction of the traditional cost.

## 3 METHODOLOGY

We introduce the **Explore-Execute Chain ($E^2C$)** framework, which decomposes reasoning tasks into two phases: Exploration and Execution. This division aims to improve reasoning efficiency, scalability, and interpretability by separating brainstorming steps from detailed calculations. As shown in Fig. 2, we first introduce a two-stage training procedure to achieve a paradigm shift and performance boost for $E^2C$ model, then we present efficient fine-tuning for specific domains and effective test-time scaling.

### 3.1 FORMAL DEFINITION OF $E^2C$

The $E^2C$ formalizes reasoning by splitting the coupled reasoning process into two conditional distributions:

$$\underbrace{p(e \mid c)}_{\text{Coupled Reasoning Process}} \quad \rightarrow \quad \underbrace{p'(\pi, e \mid c)}_{\text{Explore-Execute Chain}} \quad = \quad \underbrace{p'(\pi \mid c)}_{\text{Highly Informative}} \quad \cdot \quad \underbrace{p'(e \mid \pi, c)}_{\text{Highly Deterministic}} \tag{1}$$

The framework is defined by two core properties:

1. **(Informative Property)**. $p'(\pi \mid c)$ should be highly informative, containing the critical information necessary to solve the problem.

2. **(Deterministic Property)**. $p'(e \mid \pi, c)$ should be highly deterministic, meaning it must fully leverage the informative $\pi$.

Naturally, we semantically design $\pi$ to represent high-level strategies, while $e$ entails detailed calculations that follow $\pi$.

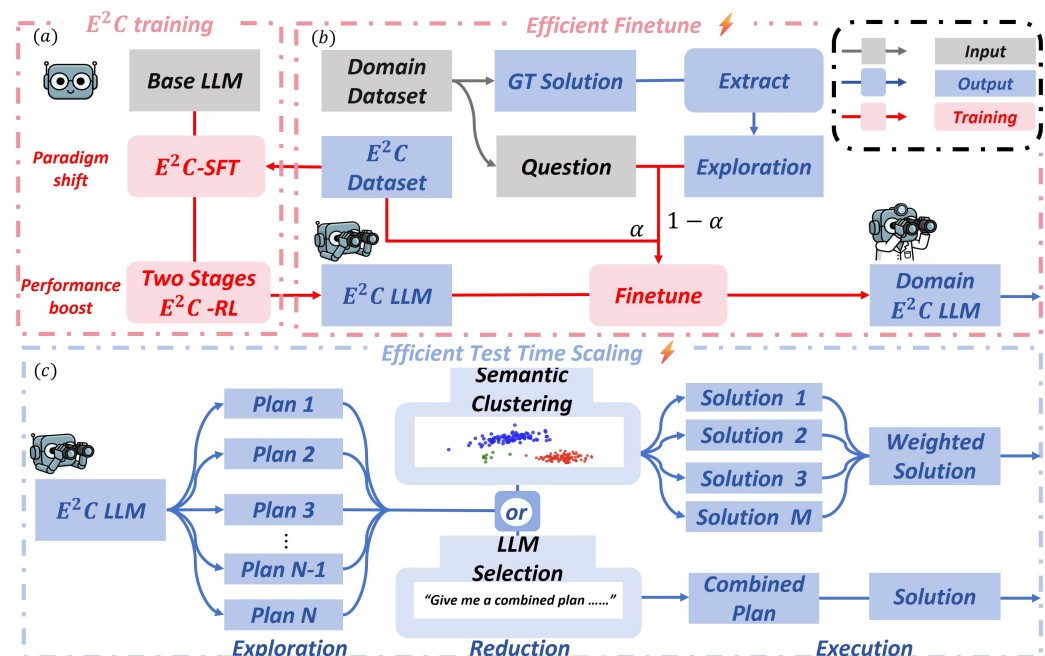

Figure 2: **Overview of E²C method.** The approach begins with E²C-SFT to achieve a paradigm shift, followed by a two-stage E²C-RL process that leverages the decomposition advantage of the new paradigm to boost performance. The resulting E²C-LLM can be efficiently adapted to new domains via EF-SFT. The exploration stage's high informativeness enables effective test-time scaling, implementable through semantic clustering or LLM selection.

## 3.2 2-STAGE TRAINING PROCEDURE: SFT AND RL

We introduce a two-stage training procedure to achieve the proposed Prop. 1 and Prop. 2. Stage 1 is Supervised Fine-Tuning (SFT), in which we construct a synthetic dataset and perform SFT to achieve a paradigm shift in reasoning and satisfy the informative Prop. 1. We do not rely solely on prompting to accomplish this paradigm transition because prompting is unstable and leads to a more significant performance drop compared to SFT training. Detailed results are presented in Tab. 1. Stage 2 employs Reinforcement Learning (RL), which incorporates a $\lambda$-coefficient on the advantage to appropriately leverage Prop. 1, thereby accelerating convergence and enhancing the determinism of execution to satisfy Prop. 2.

### 3.2.1 STAGE 1: SYNTHETIC DATASET CONSTRUCTION AND E²C-SFT

To support structured reasoning, we construct a dedicated SFT dataset through synthetic generation. A naive method is to first sample an execution trace from the base model and then summarize it into an exploration step. However, this approach is flawed: the execution is generated from $p(e \mid c)$ rather than the desired $p'(e \mid \pi, c)$, effectively hacking the causal structure. As a result, the model learns to ignore the exploration and directly mimic the base model's execution distribution, violating the intended information bottleneck.

Our method, described in Algorithm. 2, explicitly conditions the execution on the exploration. For each question, we first generate a full solution, distill it into an exploration step, and then prompt the model to produce a new execution which strictly follows the exploration. This enforces a causal dependency from exploration to execution, which is crucial for Prop. 2. The solution can also come from the ground truth. To enable a fair comparison and minimize dataset selection constraints while avoiding the introduction of extra variables, we specifically use samples from the Base LLM in our comparison experiments

### 3.2.2 STAGE 2: E²C REINFORCEMENT LEARNING (E²C-RL)

To emphasize informative reasoning, we extend hierarchical weighting (Wang et al., 2025) by assigning a higher coefficient $\lambda$ to exploration tokens, which accelerates convergence (Prop. 1), while

the entropy-reduction effect of reinforcement learning supports determinism (Prop. 2). The training objective is defined as

$$\mathcal{L}_{\text{clip}} = \frac{1}{G} \sum_{i,t} \min\Big( r_{i,t}\, \lambda_{i,t}\, \hat{A}_{i,t},\ \text{clip}(r_{i,t},\, 1 - \varepsilon,\, 1 + \varepsilon)\, \lambda_{i,t}\, \hat{A}_{i,t} \Big). \tag{2}$$

$$\mathcal{J}_{\text{GRPO}}(\theta) = \mathbb{E}[\mathcal{L}_{\text{clip}}] - \beta\, D_{\text{KL}}\big[\pi_\theta \,\|\, \pi_{\text{ref}}\big]. \tag{3}$$

where $i$ indexes the rollout in the batch, and $t$ indexes the token within that rollout, $\hat{A}_{i,t} = (r_{i,t} - \bar{r}_i)/\sigma_i$ and $r_{i,t} = r_{\text{answer}} + r_{\text{format}}$. The reward $r_{\text{answer}}$ measures answer correctness, while $r_{\text{format}}$ consists of a length reward ($r_{\text{length}}$) designed to prevent overly long and repetitive answers and an instruction reward ($r_{\text{instr}}$), quantifies the alignment between exploration and execution, ensuring that exploration trajectories approximate optimal execution strategies. The detailed description for $r_{\text{format}}$ can be found in Appendix A.2.1.

We adopt a two-stage training procedure. In the first stage, a higher temperature $\tau_1$ and larger rollout number $k_1$ are used for one epoch, encouraging broad exploration of the action space and fostering self-correction to mitigate the overly rigid adherence to the exploration plan that results. In the second stage, we reduce the temperature to $\tau_2$ and the rollout number to $k_2$, again for one epoch, and assign the advantage coefficient $\lambda_{i,t} = \lambda_{exp} > 1$ for the exploration tokens in the GRPO update. This modification explicitly prioritizes high-level reasoning in the policy gradient, thereby achieving faster and more stable convergence.

The behavior of the trained agent can be formalized by analyzing the modified GRPO objective in Eq. (3). We highlight the following quantified properties:

**1. Update emphasis: exploration vs. execution.** Let $T_{\text{exp}}$ and $T_{\text{exe}}$ be the token index sets for *exploration* and *execution*, defined by the tokens before and after the special delimiter `</EXPLORATION>` within an output $O_i = (o_{i,1}, \ldots, o_{i,|O_i|})$. The per-token policy gradient is

$$g_{i,t} \approx \lambda_{i,t}\, \hat{A}_{i,t}\, \nabla_\theta \log \pi_\theta(o_{i,t} \mid q, o_{i,<t}). \tag{4}$$

If $\lambda_{i,t} = \lambda_{\text{exp}} > 1$ for $t \in T_{\text{exp}}$ and $\lambda_{i,t} = \lambda_{\text{exe}} = 1$ for $t \in T_{\text{exe}}$, then

$$\frac{\mathbb{E}\big[\|g_{i,t}\|^2 \mid t \in T_{\text{exp}}\big]}{\mathbb{E}\big[\|g_{i,t}\|^2 \mid t \in T_{\text{exe}}\big]} \gtrsim \lambda_{\text{exp}}^2, \tag{5}$$

so exploration tokens receive significantly larger expected updates, strengthening the planning phase. The entropy dynamics are provided in Appendix A.5, which demonstrates that $\lambda_{exp}$ indeed leads to a substantial difference.

**2. Deterministic execution.** Let $o_{i,t}^\star = \arg\max_o \pi_\theta(o \mid q, o_{i,<t})$ and define the confidence margin

$$\Delta_{i,t} := \pi_\theta(o_{i,t}^\star \mid q, o_{i,<t}) - \max_{o \neq o_{i,t}^\star} \pi_\theta(o \mid q, o_{i,<t}). \tag{6}$$

Stage-2 RL (with lower temperature and fewer rollouts) increases

$$\mathbb{E}_{t \in T_{\text{exe}}}[\Delta_{i,t}] \nearrow, \qquad \mathbb{E}_{t \in T_{\text{exe}}}[H(\pi_\theta(\cdot \mid q, o_{i,<t}))] \searrow, \tag{7}$$

where $H(\pi_\theta(\cdot \mid q, o_{i,<t})) := -\sum_o \pi_\theta(o \mid q, o_{i,<t}) \log \pi_\theta(o \mid q, o_{i,<t})$ is the entropy of the token distribution at step $t$. This indicates that the execution stage becomes increasingly deterministic, with high-confidence token choices and low-variance outputs, yielding faithful and stable execution.

**3. Plan sensitivity.** Let $\hat{A}_{i,t} = \hat{A}_{i,t}^{\text{plan}}$ for $t \in T_{\text{exp}}$ be the advantage attributed to exploration tokens. Then the expected update sign satisfies

$$\mathbb{E}[\text{sgn}(g_{i,t}) \mid t \in T_{\text{exp}}] \propto \text{sgn}\Big(\mathbb{E}[\hat{A}_{i,t}^{\text{plan}}]\Big), \tag{8}$$

so high-quality plans are amplified while poor plans are suppressed.

---

**Algorithm 1** E$^2$C Test Time Scaling

---

1: Sample $K$ exploration segments: $\{e_1, e_2, \ldots, e_K\}$
2: Encode explorations to get embeddings: $V \leftarrow \{\text{Enc}(e_1), \text{Enc}(e_2), \ldots, \text{Enc}(e_K)\}$
3: **Aggregate** explorations via either:
4:   • Clustering: $E^* \leftarrow \text{Cluster-Centroids}(V)$
5:   • LLM fusion: $E^* \leftarrow \text{LLM-Aggregate}(\{e_1, \ldots, e_K\})$
6: **for** each aggregated exploration $e_i^* \in E^*$ **do**
7:     Generate execution: $a_i \leftarrow \text{Execute}(e_i^*)$
8:     Assign weight $w_i$ based on the aggregation method
9: **end for**
10: Aggregate answers: $a_{\text{final}} \leftarrow \sum w_i \cdot \delta(a_i)$
11: **return** $a_{\text{final}}$

---

### 3.3 Efficient Adaptation and Inference with E$^2$C

The modularity of our E$^2$C framework enables efficient strategies for both domain adaptation at training time and scaled aggregation at test time.

**Exploration-Focused SFT (EF-SFT).** For domain adaptation, we introduce EF-SFT. This method leverages the transferable nature of the execution component by exclusively fine-tuning on the exploration segments from domain-specific examples. These segments are mixed with the base E$^2$C dataset at a controlled ratio $\alpha$, allowing the model to efficiently learn new reasoning strategies while maintaining its core capabilities. This targeted approach significantly reduces the data and computational requirements for adaptation. A detailed algorithm can be found in the Appendix 3.

**Think Twice Before Acting: E$^2$C Test Time Scaling**. At inference time, due to the high informativeness and short length of the explorations, we can exploit this characteristic to sample a large number of plans. Afterward, using semantic clustering methods or LLMs, we select a smaller subset for execution. Specifically, we introduce two possible implementations for E$^2$C Test time scaling:

**(1) Clustering-Weighted Voting.** This approach identifies representative reasoning strategies by clustering the sampled M explorations into N clusters. Semantic similarity is measured by the cosine distance between their sentence embeddings, which are obtained from a pre-trained encoder. Only the centroid exploration from each distinct cluster proceeds to the execution phase. The final answers are aggregated using a majority vote, where the weight of each answer is proportional to its cluster size, significantly reducing redundant computations. **(2) LLM-Based Aggregation.** Alternatively, a powerful external LLM can be employed to synthesize the sampled explorations into a single, refined reasoning plan. This method consolidates key insights from multiple paths into a comprehensive exploration, which then guides a single, high-quality execution.

## 4 Experiments and Results

In this section, we describe the experimental setup of the mathematical reasoning experiment, the medical reasoning experiment, and the test-time scaling experiments. Each experiment was carried out on a single node with 8 H800 GPUs.

### 4.1 Training Protocols

We adapt our training codebase from verl (Sheng et al., 2024) and perform SFT and RL training. Our training procedures were as follows. The initial **E$^2$C-SFT** model was trained for one epoch on a 50k-sample synthetic dataset constructed from Openr1-math (deepseek, 2025) using our causal data generation algorithm (Algorithm. 2). This model was then further trained using our two-stage **E$^2$C-RL** algorithm on the DAPO-17K (Yu et al., 2025) dataset. For comparison, a baseline model was trained with the standard **GRPO** algorithm for five epochs on the same DAPO-17K data. In our domain adaptation experiments on the ReasonMed dataset, we compared a standard SFT baseline (trained on the full dataset) against our proposed **EF-SFT** method, which was trained on a targeted 50k-sample subset focused only on exploration plans, mixed with 10% regularization data.

## 4.2 EXPERIMENTS

**Mathematical Reasoning Experiment** We evaluated our E²C framework on a comprehensive suite of challenging mathematical reasoning benchmarks, including AIME'24, AIME'25, MATH500, the algebra subset of MATH (Hendrycks et al., 2024), Minerva, AMC23 and Olympiad bench (He et al., 2024) . Our proposed **E²C-(SFT+RL)** models were benchmarked against strong GRPO baselines and various ablations, with performance measured by Pass@1 accuracy averaged over 8 samples. The results demonstrate the effectiveness of our approach; for instance, on the AIME'24 benchmark, the Qwen3-4B model trained with our method achieved an accuracy of 37.5%, a significant improvement of 8.7 percentage points over the GRPO baseline.

**Medical Reasoning Experiment** To assess cross-domain generalization and data-efficient adaptation, we tested our framework on eight medical reasoning benchmarks, including MedQA (Jin et al., 2021), MedMCQA (Pal et al., 2022), and six MMLU (Hendrycks et al., 2020) subsets. We first evaluated the zero-shot transfer performance of our math-trained RL models. More critically, we compared our **EF-SFT** adaptation strategy against a standard SFT baseline. The results highlight the efficiency of E²C structure: EF-SFT improved the average accuracy of the Qwen3-8B model by 4.0 percentage points over standard SFT, while using only 10M tokens for training—less than 4% of the 286M tokens required by the baseline.

**Test-Time Scaling Experiment** A core advantage of E²C framework is its ability to facilitate highly efficient test-time scaling. We validate this superior performance-cost trade-off on the challenging AIME'2024 benchmark by comparing our methods against strong baselines, including Self-Consistency (SC) (Wang et al., 2022), Tree-of-Thoughts (ToT) (Yao et al., 2023), and the more advanced Forest-of-Thought (FoT) (Bi et al., 2025).

We evaluate two primary variants of our E²C framework, which first sample $K$ inexpensive exploration plans before committing to execution:

- **E²C-Select (Self LM-Judge)**: Uses the model itself as a judge to select the most promising plan among the $K$ samples for a single execution.
- **E²C-Select (Semantic Cluster)**: A lighter-weight alternative that embeds the $K$ plans, groups them using semantic clustering to identify representative reasoning strategies, and executes only the centroid plan from each cluster. Final answers are aggregated via a weighted majority vote based on cluster size.

To validate our design choices, we include two ablations: **E²C-SC (Self-Consistency)**, which executes all $K$ sampled plans and aggregates the final answers via majority voting to serve as a high-cost performance upper bound, and **E²C-RP** (executes one randomly selected plan). All methods are evaluated on the Qwen3-8B+E²C model across four increasing computational budget levels ($K$ or $N = 4, 8, 16, 32$).

## 4.3 RESULTS

We demonstrate our framework's reasoning capabilities in mathematical experiments, where our training process fully realizes its structural benefits. In medical reasoning, we show that the framework has stronger zero-shot generalization and validate our efficient EF-SFT method. Finally, our test-time analysis confirms that the E²C framework maintains top performance while significantly reducing computational costs.

**Mathematical Reasoning Benchmark Results** We conduct a sanity check comparing our E²C models (Qwen3-8B/4B+E²C-(SFT+RL)) against GRPO baselines, as shown in Tab. 1. Our approach outperforms the baselines by 1.5% (8B) and 1.9% (4B), validating the effectiveness of the decomposition strategy. Notably, while paradigm shifts typically risk performance degradation, our method successfully maintains and enhances model capability through careful training design. The full E²C framework ultimately surpasses the GRPO baseline by leveraging the decomposed structure, establishing a solid foundation for efficient test-time scaling.

Ablation studies in Tab. 1 reveal that E²C-RL provides significant gains over E²C-SFT+GRPO, with improvements of 3.8% (8B) and 3.2% (4B)on average accuracy, demonstrating that E²C-RL effectively exploits the decomposition advantage. Furthermore, E²C-SFT slightly outperforms the prompt-based baseline (Prompt-8B), confirming that structured training is essential for realizing the benefits of E²C paradigm.

Table 1: Performance comparison of Qwen3 models (non-thinking mode) on mathematical reasoning benchmarks. All results are reported as Pass@1 accuracy, with an 8-sample average.

| Model | AIME'24 | AIME'25 | MATH500 | Algebra | Minerva | AMC23 | Olympiad | Avg Acc | Avg Length |
|---|---|---|---|---|---|---|---|---|---|
| *Qwen3 8B Series* | | | | | | | | | |
| Qwen3-8B+GRPO (Baseline) | 36.9 | **34.4** | **88.2** | 88.2 | 33.1 | 79.3 | 60.0 | 60.0 | 1429.46 |
| Qwen3-8B+E$^2$C-SFT+GRPO | 37.5 | 32.5 | 83.5 | 86.6 | 30.8 | 76.3 | 56.8 | 57.7 | 1309.62 |
| **Qwen3-8B+E$^2$C-(SFT+RL)** | **40.6** | 33.8 | 87.7 | **90.9** | **35.8** | **80.3** | **61.3** | **61.5** | 1476.41 |
| *Qwen3 4B Series* | | | | | | | | | |
| Qwen3-4B+GRPO (Baseline) | 28.8 | **30.6** | 84.6 | 84.4 | 33.5 | 75.8 | 57.8 | 56.5 | 1263.15 |
| Qwen3-4B+E$^2$C-SFT+GRPO | 28.8 | 26.9 | 85.9 | 83.3 | 33.2 | 75.7 | 55.3 | 55.2 | 1324.18 |
| **Qwen3-4B+E$^2$C-(SFT+RL)** | **37.5** | 30.0 | **86.1** | **84.8** | **34.0** | **78.3** | **58.4** | **58.4** | 1456.34 |
| *Ablation Studies* | | | | | | | | | |
| Qwen3-8B+Prompt (Zero-shot) | 21.9 | 18.8 | 76.3 | 80.5 | 30.9 | 50.7 | 45.8 | 46.6 | 1142.38 |
| Qwen3-8B+E$^2$C-SFT | 23.1 | 21.9 | 75.8 | 80.5 | 31.5 | 51.5 | 43.2 | 46.8 | 1162.89 |

Table 2: Performance Comparison of Models with Different Training Processes: Our inference paradigm demonstrates superior generalization, while EF-SFT shows improved efficiency and robustness. The six columns from Anatomy (AN), Clinical Knowledge (CK), College Biology (CB), College Medicine (CM), Medical Genetics (MG), and Professional Medicine (PM) are validation subsets of the MMLU benchmark.

| Model | MedQA | MedMCQA | AN | CK | CB | CM | MG | PM | #Med-Tokens | Avg |
|---|---|---|---|---|---|---|---|---|---|---|
| *External Baselines* | | | | | | | | | | |
| HuatuoGPT-o1-7B | 68.4 | 57.5 | 71.9 | 78.5 | 88.2 | 67.6 | 80.0 | 77.6 | - | 73.7 |
| Baichuan-M1-14B | 76.5 | 65.2 | 77.3 | 83.6 | 87.9 | 80.7 | 89.1 | 88.8 | - | 81.1 |
| ReasonMed-7B | 66.9 | 65.1 | 75.6 | 79.3 | 79.2 | 73.4 | 85.0 | 80.9 | - | 75.7 |
| *Our Method and Ablations* | | | | | | | | | | |
| Qwen3-8B | 71.4 | 59.5 | 68.0 | 81.6 | 87.5 | 78.0 | 85.0 | 83.8 | - | 76.8 |
| Qwen3-8B + GRPO | 74.0 | 60.6 | 75.0 | 80.9 | 91.3 | 81.8 | 90.4 | **86.2** | - | 79.1 |
| **Qwen3-8B+E$^2$C-(SFT+RL)** | **74.5** | **63.1** | **77.0** | **82.2** | **92.0** | **83.0** | **92.8** | 86.0 | - | **81.1** |
| *SFT Models (using medical data)* | | | | | | | | | | |
| Qwen3-8B + standard SFT | 58.2 | 52.3 | 68.8 | 80.8 | 89.0 | 73.7 | 83.3 | 79.0 | 286M | 73.1 |
| **Qwen3-8B + E$^2$C-SFT + EF-SFT** | **65.8** | **58.2** | **72.3** | **83.8** | **89.2** | **79.7** | **87.6** | **86.2** | 10M | **77.1** |
| Llama3.1-8B + ReasonMed SFT | 42.0 | 36.8 | 45.9 | 55.4 | 61.8 | 43.2 | 38.1 | 56.9 | 286M | 47.5 |
| **Llama3.1-8B + E$^2$C-SFT + EF-SFT** | **60.3** | **53.2** | **61.8** | **69.8** | **75.9** | **64.9** | **82.0** | **72.5** | 10M | **67.5** |

**Medical Reasoning Benchmark Results** Tab. 2 presents the medical reasoning performance across three experimental settings. First, we establish competitive baselines by comparing against leading domain-specific 7B-8B models (HuatuoGPT-o1-7B (Wang et al., 2024b), ReasonMed-7B (Sun et al., 2025)) and an open-source 14B medical LLM (Baichuan-M1-14B (Bingning Wang et al., 2025)), with Qwen3-8B (Yang et al., 2025a) serving as our base model reference.

For domain adaptation, we evaluate our EF-SFT approach (Sec. 3.3) against standard SFT on both Llama3.1-8B (Dubey et al., 2024) and Qwen3-8B architectures. As shown in Tab. 2, EF-SFT achieves significant improvements of 3.9% (Qwen3-8B) and 14.5% (Llama3.1-8B) over standard SFT, while using only 3.5% of the training tokens. The zero-shot transfer results further demonstrate that our mathematically-trained RL models attain performance comparable to specialized medical LLMs, validating the strong cross-domain generalization capability of our method.

**Test-Time Scaling Performance and Efficiency Analysis** Tab. 3 demonstrates that E$^2$C framework offers a superior performance-cost trade-off. Our primary method, **E$^2$C-Select (Self LM-Judge)**, achieves a state-of-the-art 58.1% accuracy at the highest budget (K=32), surpassing baselines like Self-Consistency (54.2%). More strikingly, it reaches this performance using only **12.1k tokens**—a fraction of the cost of SC (81.6k) and FoT (128.8k) Our **E$^2$C-Select (Semantic Cluster)** variant provides an alternative trade-off. By executing the centroid of each of the main plan clusters (3 on average),it results in competitive accuracy. While its token cost is higher due to multiple executions, it remains significantly more efficient than baselines like ToT or the **E$^2$C-SC (Self-Consistency)** ablation. The high cost of E$^2$C-SC ablation validates our selective execution strategy, while the poor performance of **E$^2$C-RP (Random Plan)** underscores the necessity of an intelligent (non-random) plan selection mechanism. In summary, by efficiently scaling the inexpensive exploration phase, our framework provides a spectrum of strategies that unlock significant performance gains at a fraction of the computational cost of traditional methods.

Table 3: Test-Time Scaling Performance on AIME'2024 Benchmark with Qwen3-8B. We compare Pass@1 accuracy against the average number of generated tokens per question, demonstrating the superior performance-cost trade-off of $E^2C$ framework.

| Method | Budget Level 1 | | Budget Level 2 | | Budget Level 3 | | Budget Level 4 | |
|---|---|---|---|---|---|---|---|---|
| | Acc. (%) | Tokens (k) | Acc. (%) | Tokens (k) | Acc. (%) | Tokens (k) | Acc. (%) | Tokens (k) |
| *Standard Methods* | | | | | | | | |
| Greedy CoT ($N = 1$) | 40.6 | 2.6 | ——————— (Same as Budget Level 1) ——————— | | | | | |
| Self-Consistency | 43.9 $_{(N=4)}$ | 10.2 | 47.5 $_{(N=8)}$ | 20.4 | 51.1 $_{(N=16)}$ | 40.8 | 54.2 $_{(N=32)}$ | 81.6 |
| *Advanced Search Methods* | | | | | | | | |
| Tree-of-Thoughts (ToT) | 46.1 $_{(N=4)}$ | 14.5 | 50.2 $_{(N=8)}$ | 29.0 | 53.8 $_{(N=16)}$ | 58.0 | 56.3 $_{(N=32)}$ | 116.0 |
| Forest-of-Thought (FoT) | 48.5 $_{(N=4)}$ | 16.1 | 52.5 $_{(N=8)}$ | 32.2 | 55.8 $_{(N=16)}$ | 64.4 | 58.0 $_{(N=32)}$ | 128.8 |
| *Our Methods* | | | | | | | | |
| $E^2C$-Select (Self LM-Judge) | **49.5** $_{(K=4)}$ | **6.1** | **53.2** $_{(K=8)}$ | **7.3** | **56.4** $_{(K=16)}$ | **9.7** | **58.1** $_{(K=32)}$ | **12.1** |
| $E^2C$-Select (Semantic Cluster) | 47.8 $_{(K=4)}$ | 11.3 | 51.5 $_{(K=8)}$ | 14.8 | 54.2 $_{(K=16)}$ | 21.8 | 55.9 $_{(K=32)}$ | 35.8 |
| *Ablations* | | | | | | | | |
| $E^2C$-SC (Self-Consistency) | 50.1 $_{(K=4)}$ | 22.6 | 54.0 $_{(K=8)}$ | 45.2 | 56.9 $_{(K=16)}$ | 90.4 | 58.9 $_{(K=32)}$ | 180.8 |
| $E^2C$-RP (Random Plan) | 43.2 $_{(K=4)}$ | 6.1 | 44.5 $_{(K=8)}$ | 7.3 | 45.1 $_{(K=16)}$ | 9.7 | 45.8 $_{(K=32)}$ | 12.1 |

**Ablations and Analysis** Our ablation studies validate our key design choices. As shown in Part A of Tab. 4, our causal data generation strategy (Algorithm 1) is essential, achieving near-perfect plan adherence (0.998) that is critical for $E^2C$ paradigm. Part B demonstrates the framework's efficiency in domain adaptation; performance on medical benchmarks peaks after a brief training period (300 iterations, nearly 5k samples) and declines thereafter, highlighting the data-efficient nature of fine-tuning only the exploration phase. Part C shows that incorporating a small proportion of regularization data ($\alpha = 10\%$) is superior to both using no regularization ($\alpha = 0\%$) and training on the full exploration-execution sequence ($\alpha = 100\%$), highlighting the efficiency and robustness derived from

Table 4: An ablation analysis shows the validity of our data construction methodology by quantifying plan adherence (top), identifies the optimal training iteration count for medical domain SFT (middle), and shows the impact of data mixing (bottom).

Part A: Ablation on SFT Data Construction Strategy

| SFT Data Strategy | Plan-Guided | Plan-Free |
|---|---|---|
| Flawed (Reverse-Causal Summary) | 0.499 | 0.864 |
| **Proposed (Causal Generation)** | **0.998** | **1.000** |

Part B: Ablation on Exploration-Focused SFT Training Steps

| Training Steps | GSM8K | Anatomy | CK | MedQA | MATH |
|---|---|---|---|---|---|
| SFT + 100 iter | 83.0 | 70.4 | 80.6 | **66.4** | 70.5 |
| **SFT + 300 iter** | 83.3 | **72.3** | **83.8** | 66.1 | **71.2** |
| SFT + 2000 iter | **84.0** | 69.8 | 78.4 | 63.5 | 71.1 |

Part C: Ablation on Regularization Data

| | GSM8K | Anatomy | CK | MedQA | MATH |
|---|---|---|---|---|---|
| Medical Reg. ($\alpha$=10%) | 82.1 | **74.4** | 79.5 | 66.8 | 69.5 |
| **Math Reg. ($\alpha$=10%)** | **83.3** | 72.3 | **83.8** | 66.1 | **71.2** |
| Medical Reg. ($\alpha$=100%) | 80.1 | 70.0 | 81.5 | **67.0** | 65.8 |
| No Reg. ($\alpha = 0\%$) | 82.1 | 68.2 | 77.8 | 63.5 | 70.8 |

the exploration-focused approach. Additionally, it suggests that using regularization data from the base $E^2C$-SFT dataset (i.e., using Math as Regularization) is more effective than using domain-specific medical data for regularization, indicating that there is no need to generate regularization data for the specific target domain.

## 5 LIMITATIONS AND FUTURE WORK

The $E^2C$ framework, while demonstrating advanced reasoning capabilities, currently faces limitations in supporting long-chain reasoning models such as gpt-o1 (OpenAI, 2024) and deepseek-r1 (Guo et al., 2025) due to architectural differences. To address this, we plan to develop multi-round exploration and execution mechanisms that enable iterative refinement and more effective decomposition of complex, long-horizon tasks.

At the same time, the decoupled nature of $E^2C$ offers unique advantages for human-AI collaboration. The exploration phase provides users with immediate visibility into the model's reasoning process, facilitating rapid feedback and collaborative ideation. The execution phase serves as a transparent and reliable module that translates high-level plans into actionable results, significantly enhancing the interpretability, controllability, and usability of the system. We believe these characteristics establish a foundation for more adaptive and user-centered AI assistants, with strong potential to support human-in-the-loop applications requiring complex reasoning and interactive decision-making.

## 6 CONCLUSION

Through the proposed Explore-Execute Chain (E$^2$C), we introduce a novel reasoning framework that decouples exploration from execution, enhancing both efficiency and interpretability. Our two-stage SFT+RL training approach, supported by a dedicated data construction method and token-specific reward scaling, enables faithful plan adherence and robust paradigm transition. The framework effectively concentrates information in exploration, allowing domain adaptation using only 3.5% of training tokens and achieving a superior performance-cost trade-off on complex reasoning benchmarks compared to strong baselines. This also opens up new avenues for users to interact with reasoning models.

ETHICS STATEMENT

This work studies a reasoning framework, the **Explore–Execute Chain ($E^2C$)**, which separates lightweight exploratory sketches from a final execution step to improve efficiency, transparency, and controllability of LLM reasoning. Our experiments fine-tune and evaluate general-purpose LLMs on publicly available benchmarks (e.g., mathematics and domain reasoning datasets). We do not collect new human data, do not involve human or animal subjects, and do not process personally identifiable or sensitive information. Any third-party datasets used in this paper are publicly released for research purposes by their respective providers; we follow their licenses and usage terms. $E^2C$ paradigm increases interpretability by exposing intermediate "exploration" traces, which can facilitate auditing and discourage over-reliance on hidden chain-of-thought. This study complies with the conference's Code of Ethics.

REPRODUCIBILITY STATEMENT

We have made extensive efforts to ensure the reproducibility of our work. All code used in this paper will be publicly released to facilitate independent verification and further research. We describe our experimental setup in Sec. 4.1. Detailed hyperparameters for training $E^2C$-SFT, $E^2C$-RL, GRPO, and EF-SFT are provided in Appendix A.2.1. Detailed setup for TTS experiment can be found in Appendix A.4. We also include the prompt templates for data generation, the zero-shot prompt model, and $E^2C$-Select (Self LM-Judge) in Appendix A.6.

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

## USE OF LARGE LANGUAGE MODELS

We utilized a large language model to enhance the language and clarity of our manuscript. Specifically, we employed Gemini 2.5 flash with the following prompt to refine the initial draft: *I am writing an academic paper in English. Please polish the following draft so that it adheres to the conventions of academic writing.*

## A APPENDIX

### A.1 COGNITIVE MODEL ANALYSIS

In this section, we analyze cognitive models to derive high-level design insights for our method.

### A.1.1 RUBICON MODEL OF ACTION PHASES

The **Rubicon Model of Action Phases** (Achtziger & Gollwitzer, 2007), proposed by **Heckhausen** and **Gollwitzer**, provides a framework for how individuals prepare for and pursue goals. It divides goal pursuit into four stages: **goal setting**, **planning**, **action**, and **evaluation**.

1. **Goal Setting**: Individuals identify and adopt a goal, motivated by a need or desire.
2. **Planning**: After a goal is adopted, individuals generate strategies to achieve it and assess their potential effectiveness.
3. **Action**: Once a strategy is selected, the individual commits to it and executes it. Crossing the "Rubicon" marks this commitment and the transition to action.
4. **Evaluation**: Outcomes are assessed to inform adjustments to subsequent plans or actions.

A key contribution of the Rubicon Model is the sharp distinction between planning and execution. After commitment (crossing the Rubicon), attention is devoted to execution rather than continued exploration or second-guessing. This separation mitigates cognitive overload that could arise from ongoing re-evaluation during task execution.

### A.1.2 CONNECTING E$^2$C WITH RUBICON MODEL

We formally express E$^2$C as

$$\underbrace{p(e \mid c)}_{\text{Coupled Reasoning Process}} \rightarrow \underbrace{p'(\pi, e \mid c)}_{\text{Explore-Execute Chain}} = \underbrace{p'(\pi \mid c)}_{\color{red}\text{Highly Informative}} \cdot \underbrace{p'(e \mid \pi, c)}_{\color{blue}\text{Highly Deterministic}} \tag{9}$$

$p'(\pi \mid c)$ **as the Planning Phase**: In the Rubicon framework, planning entails generating candidate strategies. Analogously, in E$^2$C, $p'(\pi \mid c)$ produces multiple candidate plans $\pi$ from context $c$. These plans are highly informative, capturing the critical information needed to solve the task. This exploration corresponds to the goal-setting and planning stages, where alternatives are considered before selection.

$p'(e \mid \pi, c)$ **as the Execution Phase**: Once plans are available, E$^2$C transitions to execution. The distribution $p'(e \mid \pi, c)$ reflects a highly deterministic process that follows the selected plan $\pi$ under context $c$. This mirrors the action phase of the Rubicon Model: the agent executes the committed plan without revisiting discarded alternatives.

Thus, the separation between $p'(\pi \mid c)$ and $p'(e \mid \pi, c)$ in E$^2$C parallels the explore–then–execute dynamics of the Rubicon Model: first enumerate options, then execute deterministically.

### A.1.3 COGNITIVE AND COMPUTATIONAL EFFICIENCY

Separating exploration from execution confers efficiency benefits in both cognition and computation. Cognitively, once commitment occurs, resources are focused on carrying out the chosen plan without distraction from alternatives. Computationally, E$^2$C avoids the overhead of re-evaluating multiple plans during execution. The deterministic execution phase concentrates compute on following the selected plan, yielding faster and more reliable performance than continually interleaving exploration with action.

### A.1.4 INTERPRETABILITY AND TRANSPARENCY

The exploration–execution split also improves **interpretability**. In the Rubicon Model, one can explain an action by the plan selected during the planning stage. Likewise, E$^2$C makes the reasoning path explicit: multiple candidate plans are generated (exploration), and one is chosen and followed (execution). This transparency further supports **scalability**: the exploration component can be adapted to new tasks and domains, while the execution component remains stable, enabling flexible and extensible reasoning across settings.

### A.2 THE DETAILS OF THE EXPERIMENTS

In this section, we introduce the details of our main experiments in the main paper for reproducibility purposes, including the detailed hyperparameter settings and the reward designs.

### A.2.1 HYPERPARAMETER SETTINGS

**E$^2$C-SFT and EF-SFT Training** For both E$^2$C-SFT and EF-SFT training, the hyperparameters are summarized in Tab. 5:

| Hyperparameter | Value |
|---|---|
| Learning Rate | $1.0 \times 10^{-5}$ |
| Optimizer | Adam(Kingma, 2014) ($\beta_1 = 0.9, \beta_2 = 0.95$) |
| Weight Decay | 0.01 |
| Learning Rate Scheduler | Cosine with 10% warmup ratio |
| Batch Size | 160 |
| Micro-batch Size per GPU | 20 |
| Gradient Clipping | 1.0 |
| Total Epochs | 1 |

Table 5: Hyperparameters for E$^2$C-SFT and EF-SFT Training

**E$^2$C-RL and GRPO Training** The hyperparameters for E$^2$C-RL and GRPO training are summarized in Tab. 6, where the experiments include E$^2$C Stage 1 (E2C-stg1), E$^2$C Stage 2 (E2C-stg2), and GRPO:

| Hyperparameter | E$^2$C-stage1 | E$^2$C-stage2 | GRPO |
|---|---|---|---|
| Batch Size | 256 | 256 | 128 |
| Overlong Buffer Length | 4096 | 4096 | 4096 |
| Maximum Response Length | 8192 | 8192 | 8192 |
| Learning Rate | $1.0 \times 10^{-6}$ | $1.0 \times 10^{-6}$ | $1.0 \times 10^{-6}$ |
| Mini-batch Size for GRPO Updates | 32 | 32 | 32 |
| KL Loss Coefficient $\beta$ | 0.001 | 0 | 0 |
| Rollout Number $k$ | 32 | 8 | 8 |
| Temperature | 1.3 | 1.0 | 1.0 |
| Training Epochs | 1 | 1 | 5 |
| Clip ratio ($\varepsilon$) | 0.2 | 0.2 | 0.2 |

Table 6: Hyperparameters for E$^2$C-RL and GRPO Training

### A.2.2 REWARD DETAILS FOR RL TRAINING

**Format Reward Calculation for E$^2$C Training** For the **E$^2$C** training, the format reward consists of two components: the length reward and the instruction reward. These rewards are computed as follows:

**Length Reward**: This reward measures how well the output length matches the expected length. It is computed as:

$$r_l = -\text{clip}\left(0, 1, \frac{L - L_{valid}}{L_{buffer}}\right)$$

where: $L$ is the length of the generated output; $L_{valid}$ is the length of the valid portion of the response; $L_{buffer}$ is the overlong buffer length.

**Instruction Reward**: The instruction reward is specific to the **E$^2$C** model and is added to the reward function when it comes to E$^2$C model. This reward measures the alignment between the instructions generated during the exploration phase and the execution phase. It is computed by extracting the step titles from both the exploration and execution phases using regular expressions. Denote these sets of instructions as $S_1$ (exploration) and $S_2$ (execution). The instruction reward is defined as:

$$r_{\text{instr}} = 0.1 * \left(\frac{|S_1 \cap S_2|}{\max(|S_1|, |S_2|)} - 1\right)$$

where: $S_1$ is the set of instructions generated during the exploration phase; $S_2$ is the set of instructions generated during the execution phase; $|S_1 \cap S_2|$ is the intersection of the sets $S_1$ and $S_2$; $\max(|S_1|, |S_2|)$ is the maximum size of the two sets.

The instruction reward incentivizes the model to generate instructions that align well between the exploration and execution phases, encouraging consistency. This reward is crucial for $\mathbf{E^2C}$ models to ensure that the reasoning process is coherent between the exploration and execution stages.

**Format Reward Calculation for GRPO Training**  For **GRPO** training, the format reward is simpler and consists solely of the length reward, which is computed using the same formula as in $\mathrm{E^2C}$:

$$
r_l = -\mathrm{clip}\left(0, 1, \frac{\mathrm{length_{output}} - \mathrm{valid_{length}}}{\mathrm{buffer_{length}}}\right)
$$

In GRPO, no instruction reward is applied, and the focus is entirely on the length of the response, ensuring that the output adheres to the expected length constraints.

## A.3  DETAILS OF THE ALGORITHM

**Algorithm of $\mathbf{E^2C}$-SFT Data Generation**  Algorithm 2 is a formal and detailed description for $\mathrm{E^2C}$-SFT Data Generation.

---

**Algorithm 2** $\mathrm{E^2C}$-SFT Data Generation

---

1: $\mathcal{D}_{\mathrm{synth}} \leftarrow \emptyset$
2: **for** each question $q$ **do**
3:     solution $\leftarrow \mathrm{Model_{base}}(q)$
4:     exploration $\leftarrow \mathrm{Summarize(solution)}$
5:     prompt $\leftarrow$ "Given question: $q$. Follow exploration: exploration. Execute step-by-step:"
6:     execution $\leftarrow \mathrm{Model_{base}(prompt)}$
7:     $\mathcal{D}_{\mathrm{synth}} \leftarrow \mathcal{D}_{\mathrm{synth}} \cup \{(q, (\mathrm{exploration, execution}))\}$
8: **end for**
9: **return** $\mathcal{D}_{\mathrm{synth}}$

---

**Algorithm of Exploration-Focused SFT (EF-SFT) Data Generation**  Algorithm 3 is a formal and detailed description for EF-SFT Data Generation.

---

**Algorithm 3** EF-SFT Data Generation

---

**Require:** Base $\mathrm{E^2C}$ dataset $\mathcal{D}_{\mathrm{base}}$
**Require:** Domain-specific dataset $\mathcal{D}_{\mathrm{domain}}$
**Require:** Mixing ratio $\alpha \in [0, 1]$, Target Dataset size $n_{\mathrm{target}}$
**Ensure:** EF-SFT training dataset $\mathcal{D}_{\mathrm{EF\text{-}SFT}}$
1: $\mathcal{D}_{\mathrm{explore}} \leftarrow \emptyset$
2: **for** each example $(q, a) \in \mathcal{D}_{\mathrm{domain}}$ **do**
3:     Extract exploration segment: $e \leftarrow \mathrm{ExtractExploration}(a)$
4:     $\mathcal{D}_{\mathrm{explore}} \leftarrow \mathcal{D}_{\mathrm{explore}} \cup \{(q, e)\}$
5: **end for**
6: $n_{\mathrm{base}} \leftarrow \alpha \times n_{\mathrm{target}}$                               $\triangleright \alpha\%$ from base dataset
7: $n_{\mathrm{explore}} \leftarrow (1 - \alpha) \times n_{\mathrm{target}}$                  $\triangleright (1 - \alpha)\%$ from exploration data
8: $\mathcal{D}_{\mathrm{base}}^{\mathrm{sub}} \leftarrow \mathrm{Subsample}(\mathcal{D}_{\mathrm{base}}, n_{\mathrm{base}})$
9: $\mathcal{D}_{\mathrm{explore}}^{\mathrm{sub}} \leftarrow \mathrm{Subsample}(\mathcal{D}_{\mathrm{explore}}, n_{\mathrm{explore}})$
10: $\mathcal{D}_{\mathrm{EF\text{-}SFT}} \leftarrow \mathcal{D}_{\mathrm{base}}^{\mathrm{sub}} \cup \mathcal{D}_{\mathrm{explore}}^{\mathrm{sub}}$
11: **return** $\mathcal{D}_{\mathrm{EF\text{-}SFT}}$

---

A.4   TEST-TIME SCALING EXPERIMENTAL DETAILS

This section provides a detailed description of the experimental setup for the test-time scaling comparison presented in Table 3, ensuring reproducibility.

**Objective and General Setup**   The primary goal was to evaluate the performance-cost trade-off of our $E^2C$ framework against established baselines on the AIME'2024 benchmark. All methods were evaluated using the same checkpoint: the **Qwen3-8B+$E^2C$-(SFT+RL)** model. This ensures a fair comparison of the inference strategies themselves, rather than the underlying models. For all generative steps that require diversity (e.g., sampling paths or plans), a temperature of 0.9 was used. Performance is reported as Pass@1 accuracy, and cost is measured by the average total number of tokens generated per question.

**Baseline Methods**

- **Greedy CoT**: A single reasoning chain was generated for each question using greedy decoding (N=1). This serves as the most basic baseline.
- **Self-Consistency (SC)**: For each budget level $N \in \{4, 8, 16, 32\}$, we generated N full, independent CoT reasoning chains. The final answer was determined by a majority vote among the N outputs.
- **Tree-of-Thoughts (ToT) & Forest-of-Thought (FoT)**: We implemented these advanced search methods following the standard procedures described in their respective papers (Yao et al., 2023; Bi et al., 2025). The number of reasoning paths explored was set to match the budget levels $N \in \{4, 8, 16, 32\}$ to ensure a comparable computational scale.

**$E^2C$ Methods and Ablations**   All $E^2C$ variants begin by sampling $K \in \{4, 8, 16, 32\}$ exploration plans from the same model. The subsequent steps differ as follows:

- **$E^2C$-Select (Self LM-Judge)**: The K sampled plans and the original question were formatted into a prompt for the model to act as a judge and select the single most promising plan. A single execution was then generated conditioned on this selected plan.
- **$E^2C$-Select (Semantic Cluster)**: This method involves a multi-step, voting-based process: (1) Each of the K plans was embedded into a vector using the standard `all-mpnet-base-v2` sentence-transformer model. (2) We applied K-Means clustering to group these embeddings into M=3 distinct clusters. (3) The plan closest to the centroid of each of the M clusters was selected for execution, resulting in M executions. (4) The final answer was determined by a weighted majority vote over the M outcomes, where each vote's weight was proportional to the size of its corresponding cluster.
- **$E^2C$-SC (Self-Consistency)**: This ablation executed all K sampled plans independently. The final answer was determined by a standard majority vote over the K resulting outcomes. This serves as a high-cost upper bound for the $E^2C$ paradigm.
- **$E^2C$-RP (Random Plan)**: As a simple ablation, one plan was randomly selected from the K samples and then executed to produce a single answer.

A.5   ENTROPY VISUALIZATION OF DIFFERENT RL SETTINGS AND ANALYSIS

In this part, we visualize the entropy dynamics and the accuracy on the AIME'24 benchmark during RL training. The results demonstrate that applying our token-weighting coefficient $\lambda_{i,t}$ to exploration tokens facilitates a rapid drop in entropy and a better performance improvement, as shown in **Fig. 3**. This is achieved by effectively amplifying high-quality plans while suppressing poor ones.

A.6   PROMPT DETAILS

**$E^2C$-SFT Dataset Construction prompt**

EXPLORATION PHASE PROMPT   The following prompt is used to extract the high-level exploration plan from the reasoning process:

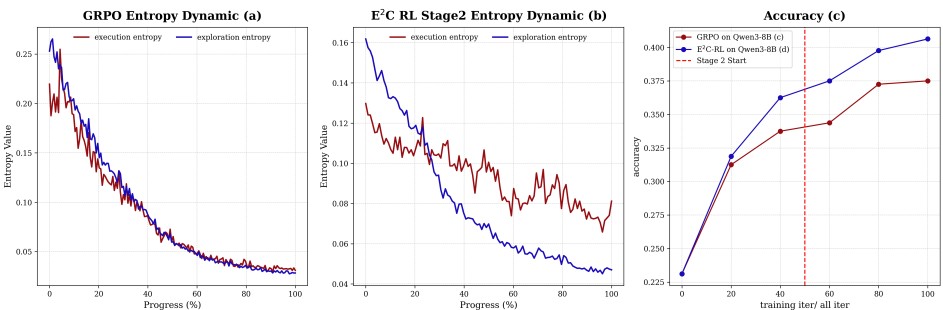

Figure 3: A comparison of training dynamics on the AIME'24 benchmark. The application of our token-weighting coefficient $\lambda_{i,t}$ (b) facilitates faster entropy reduction and superior performance improvement compared to the baseline without it (a).

**Role:** You are an expert problem-solver.
**Task:** Distill a complex reasoning process into a clear, actionable plan.

**Input:**

- **Problem:** `<question>`
- **Reasoning Process:** `<content>`

**Output Requirements:**

1. **Format:** Present the summary as a numbered list (e.g., 1., 2., 3.).
2. **Content:** For each step, describe only the essential action to be taken (e.g., "Calculate X," "Verify Y"). Be concise and prescriptive.
3. **Focus:** Omit explanations, justifications, or intermediate conclusions.

**Goal:** Create a high-level plan that is easy to follow and execute.

EXECUTION PHASE PROMPT  The following prompt is used to generate the detailed execution steps based on the exploration plan:

**Role:** You are a meticulous problem solver.
**Task:** Solve the given question by strictly following the provided guideline, showing all detailed reasoning.

**Input:**

- **Question:** `<question>`
- **Guideline:** `<content>`

**Output Requirements:**

1. Follow the guideline exactly, numbering each step accordingly (e.g., 1., 2., ...).
2. Do not include any content outside the solution steps.
3. Begin from Step 1, expanding each step with necessary calculations and logical reasoning.
4. Conclude by placing the final answer within a '`\boxed{}`' environment.

**Important:** Ensure every mathematical or logical operation is explicitly shown.

**EF-SFT dataset Construction prompt**  The following prompt is used to extract the exploration part for EF-SFT dataset in medical domain.

**Role:** You are a professional doctor.
**Task:** Summarize the diagnostic reasoning process into a concise, actionable

guideline.

**Input:**

- **Question:** `<question>`
- **Reasoning Process:** `<content>`

**Output Requirements:**

1. **Structure:** Present the summary as a numbered list (1., 2., ...), starting directly with the first step.
2. **Conciseness:** Use no more than 5 steps. Each step must be under 15 words and state only the critical objective (e.g., "Assess cardiac function").
3. **Focus:** Highlight the most critical diagnostic step. Omit all explanations, justifications, or unrelated content.

**Goal:** Create a concise and accurate diagnostic plan focused on key actions.

**LLM-Combination Prompt** To enable the model to select the most promising exploration plan, we use the following prompt. The model is instructed to act as an impartial judge, evaluating the provided plans based on their clarity, correctness, and likelihood of leading to a successful solution.

**Role:** You are an expert mathematical reasoner and an impartial judge. Your task is to evaluate several proposed plans for solving a given math problem and identify the single best one.

**Input:**

- **Problem:** `<problem>`
- **Candidate Plans:** A numbered list of K exploration plans. Plan 1: $<exploration_1 >$ $Plan2 :< exploration_2 > ...PlanK :< exploration_K >$

**Instructions:**

1. Carefully analyze the problem and each of the K candidate plans.
2. Assess the plans based on their logical soundness, potential for success, and efficiency.
3. Select the single best plan that is most likely to lead to a correct and complete solution.

**Output Format:** Output only the full text of the single best plan you have selected. Do not add any extra commentary, explanation, or formatting.

**Adherence Judge Prompt** The following prompt is used to evaluate whether an execution strictly adheres to the provided exploration plan.

**Role:** You are a rigorous evaluator. Your task is to judge if the execution strictly follows the exploration plan.

**Input:**

- **Question:** question
- **Exploration (PLAN):** exploration
- **Execution:** execution

**Evaluation Criteria:** Provide a final score of 0, 0.5, or 1.0 based on the following:

- **1.0 (Strictly Adheres):** The execution follows the exploration's logic and key steps exactly. All critical reasoning or calculations in the exploration are present and correctly implemented in the execution.
- **0.5 (Partially Adheres):** The execution addresses the main goal but deviates in specifics: it may skip non-critical steps, change the order of minor steps, or contain minor logical gaps while reaching a correct conclusion.
- **0.0 (Does Not Adhere):** The execution ignores major exploration steps, follows a different approach, contains significant logical errors, or fails to implement the exploration's core intent.

**Output Format:** First, provide a concise rationale comparing the exploration and execution. Then, output the score strictly as:
```
[RATIONALE]: <your analysis>
[SCORE]: <0.0, 0.5, or 1.0>
```

## A.7   E$^2$C REASONING EXAMPLES

### Example 1(Math, from AMC23): Meeting Problem

INPUT QUESTION    Cities $A$ and $B$ are $45$ miles apart. Alicia lives in $A$ and Beth lives in $B$. Alicia bikes towards $B$ at 18 miles per hour. Leaving at the same time, Beth bikes toward $A$ at 12 miles per hour. How many miles from City $A$ will they be when they meet? Provide the final answer in the boxed.

EXPLORATION PHASE    1. Identify the speeds and initial distance between the two cities. 2. Determine the combined speed at which the distance between Alicia and Beth is closing. 3. Calculate the time it takes for them to meet by dividing the initial distance by their combined speed. 4. Use the time to calculate the distance Alicia travels from City A.

EXECUTION PHASE

1. **Identify the speeds and initial distance between the two cities.** Distance between City A and City B: 45 miles. Alicia's speed: 18 miles per hour (towards B). Beth's speed: 12 miles per hour (towards A)

2. **Determine the combined speed at which the distance between Alicia and Beth is closing.** Since they are moving towards each other, their speeds add up:

$$\text{Combined speed} = 18 + 12 = 30 \text{ miles per hour}$$

3. **Calculate the time it takes for them to meet by dividing the initial distance by their combined speed.** The time until they meet is:

$$\text{Time} = \frac{\text{Distance}}{\text{Combined speed}} = \frac{45}{30} = 1.5 \text{ hours}$$

4. **Use the time to calculate the distance Alicia travels from City A.** Alicia's distance from City A when they meet is:

$$\text{Distance} = \text{Speed} \times \text{Time} = 18 \times 1.5 = 27 \text{ miles}$$

FINAL ANSWER

$$\boxed{27}$$

### Example 2 (Math, from AIME'24): Table Tennis Tournament Problem

INPUT QUESTION    In a table tennis tournament every participant played every other participant exactly once. Although there were twice as many right-handed players as left-handed players, the number of games won by left-handed players was $40\%$ more than the number of games won by right-handed players. (There were no ties and no ambidextrous players.) What is the total number of games played? Provide the final answer in the boxed.

EXPLORATION PHASE    1. Define variables for the number of left-handed and right-handed players. 2. Express the total number of games played in terms of the total number of players. 3. Determine the total number of games won by left-handed and right-handed players. 4. Set up an equation based on the given relationship between wins. 5. Solve the equation to find the total number of players. 6. Calculate the total number of games played.

EXECUTION PHASE

1. **Define variables for the number of left-handed and right-handed players.** Let $L$ be the number of left-handed players, and $R$ be the number of right-handed players. Given: $R = 2L$

2. **Express the total number of games played in terms of the total number of players.** Total players: $L + R = 3L$ Total games: $\binom{3L}{2} = \frac{3L(3L-1)}{2}$

3. **Determine the total number of games won by left-handed and right-handed players.** Let $W_L$ be wins by left-handed players, $W_R$ be wins by right-handed players. Total wins: $W_L + W_R = \frac{3L(3L-1)}{2}$ Given: $W_L = 1.4W_R$

4. **Set up an equation based on the given relationship between wins.** Substitute $W_L = 1.4W_R$ into total wins: $1.4W_R + W_R = \frac{3L(3L-1)}{2}$ $2.4W_R = \frac{3L(3L-1)}{2}$

5. **Solve the equation to find the total number of players.** Try $L = 3$: $W_R = \frac{3 \times 3 \times (9-1)}{4.8} = \frac{72}{4.8} = 15$ $W_L = 1.4 \times 15 = 21$ Total wins: $15 + 21 = 36$

6. **Calculate the total number of games played using the total number of players.** Total games: $\frac{3 \times 3 \times (9-1)}{2} = \frac{72}{2} = 36$

FINAL ANSWER

$$\boxed{36}$$

**Example 3 (MedicalFrom MMLU-Anatomy): Urinary System Question**

INPUT QUESTION   Which of the following best describes the structure that collects urine in the body? A. Bladder B. Kidney C. Ureter D. Urethra Please reasoning step-by-step. Provide the final answer in the boxed.

EXPLORATION PHASE   1. Identify the function of each organ listed in the options. 2. Determine which organ is responsible for storing urine. 3. Eliminate options that do not fit the function of collecting urine. 4. Select the organ that best matches the description of collecting urine.

EXECUTION PHASE

1. **Identify the function of each organ listed in the options:**

   - **A. Bladder:** Hollow, muscular organ that stores urine until excretion
   - **B. Kidney:** Filters blood and produces urine, but does not store urine
   - **C. Ureter:** Transports urine from kidneys to bladder, does not store urine
   - **D. Urethra:** Carries urine from bladder to outside body, does not store urine

2. **Determine which organ is responsible for storing urine:** The bladder is the primary organ responsible for urine storage. It expands as it fills with urine and contracts during urination.

3. **Eliminate options that do not fit the function of collecting urine:**

   - Kidney (B): Produces urine but does not store it
   - Ureter (C): Transports urine but does not store it
   - Urethra (D): Excretes urine but does not store it
   - Bladder (A): Only organ designed for urine collection and storage

4. **Select the organ that best matches the description of collecting urine:** The bladder is the structure that collects and stores urine in the body.

FINAL ANSWER

$$\boxed{A}$$

**Example 4 (Test-Time Scaling, from AIME'24): Table Tennis Tournament Problem**  This example demonstrates the application of our two primary test-time scaling methods using the same input question from Example 2.

INPUT QUESTION  In a table tennis tournament every participant played every other participant exactly once. Although there were twice as many right-handed players as left-handed players, the number of games won by left-handed players was $40\%$ more than the number of games won by right-handed players. (There were no ties and no ambidextrous players.) What is the total number of games played? Provide the final answer in the boxed.

STEP 1: SAMPLE K EXPLORATION PLANS (K=4)  First, the $E^2C$ model is prompted to generate K=4 diverse, high-level exploration plans for the problem.

- **Plan 1** 1. Let L be the number of left-handed players, the total games are C(3L, 2). 2. Let W_L be wins by lefties and W_R be wins by righties. 3. Solve this system for L, then find the total games.

- **Plan 2** 1. The total number of players must be a multiple of 3, let's call it 3L. The total games played is C(3L, 2). 2. Let's test small integer values for L (L=1, 2, 3...) and check if the resulting total games can be split into wins for left- and right-handed players satisfying the 40% more condition.

- **Plan 3** 1. Assume the number of wins is proportional to the number of players. Let right-handed players have W_R wins. 2. Left-handed players have half the number of players, so they should have W_L wins. 3. Set up W_L = 1.4 * W_R and solve based on the total number of games.

- **Plan 4** 1. Let the number of wins by right-handed players be W_R. Then the wins by left-handed players is 1.4 * W_R. 2. The total number of games is 2.4 * W_R. The total number of games is also given by C(3L, 2). 3. Set C(3L, 2) = 2.4 * W_R and find an integer solution for L.

METHOD A: $E^2C$-SELECT (SELF LM-JUDGE)  The four plans above, along with the original question, are fed into the model with the Self LM-Judge prompt. The model evaluates the plans and selects the most robust and direct strategy.

1. **Selection:** The Self LM-Judge identifies Plan 1 as the most comprehensive and logically sound approach, as it correctly sets up the system of equations from first principles.

2. **Execution:** A single execution is performed, conditioned only on Plan 1. This execution proceeds exactly as detailed in Example 2, arriving at the correct answer.

**Final Answer (Self LM-Judge):** $\boxed{36}$

METHOD B: $E^2C$-SELECT (SEMANTIC CLUSTER)  This algorithmic method clusters the plans before execution.

1. **Embedding and Clustering:** The four plans are embedded into vectors. A clustering algorithm (e.g., K-Means) is applied and identifies M=3 distinct strategic groups:
   - **Cluster A** Plan 1 and Plan 4 are grouped together as they both use a correct algebraic formulation. (Cluster Size = 2)
   - **Cluster B** Plan 2 is identified as a distinct trial-and-error strategy. (Cluster Size = 1)
   - **Cluster C** Plan 3 is isolated as it is based on an incorrect assumption. (Cluster Size = 1)

2. **Centroid Execution:** The plan closest to the centroid of each cluster is selected and executed.
   - **Execution of A (from Plan 1):** Results in the correct answer, **36**.
   - **Execution of B (from Plan 2):** Also results in the correct answer, **36**.
   - **Execution of C (from Plan 3):** The flawed logic leads to an incorrect answer, e.g., **45**.

3. **Weighted Majority Vote:** The final answer is determined by a weighted vote of the execution outcomes.

   - Vote for answer "36": Received from Cluster A (weight=2) and Cluster B (weight=1). Total weight = $2 + 1 = 3$.
   - Vote for answer "45": Received from Cluster C (weight=1). Total weight = $1$.

   The answer "36" has the highest weight.

**Final Answer (Semantic Cluster):** 36

### A.8 PURE PROMPT-BASED E$^2$C

We product an experiment with pure prompt-based E$^2$C on Qwen3-8B. For each problem we first sample $K$ independent *exploration* traces by prompting the model $K$ times with a short exploration prompt; each exploration is a concise (2–4 short sentence) reasoning sketch that does not contain the final answer. We then combine the $K$ explorations into a single execution prompt (providing the problem and the numbered explorations) and ask the model to produce one final *Execution:* section that computes the final answer. Performance is reported as pass@5 for different values of $K$. The results are much worse than the E$^2$C model with E$^2$C-(SFT+RL), which demonstates that a prompt engeneering is not enough.

**Exploration prompt** The following prompt was used to generate each individual exploration (one exploration per model call).

> **Role:** You are a careful math problem solver.
> **Input:**
> - **Problem:** `<problem>`
>
> **Instructions:**
> - Produce exactly one short reasoning sketch (an *exploration*) that helps approach the problem.
> - The exploration must be concise (about 2–4 short sentences).
> - Do **not** produce the final answer in this call.
> - Stop immediately after the single exploration text and do not append any extra commentary, labels, or formatting.
>
> **Output format:** A single short exploration paragraph (2–4 short sentences) and nothing else.

**Execution prompt** The following prompt was used to synthesize the $K$ independently sampled explorations into a final execution.

> **Role:** You are a careful math problem solver.
> **Input:**
> - **Problem:** `<problem>`
> - **Explorations:**
>   Exploration 1: `<exploration 1>`
>   Exploration 2: `<exploration 2>`
>   ⋮
>   Exploration $\{K\}$: `<exploration K>`

Table 7: Pass@5 accuracy (%) for different numbers of sampled explorations $K$.

| Dataset | $K = 2$ | $K = 3$ | $K = 4$ | $K = 5$ |
|---------|---------|---------|---------|---------|
| MATH500 | 84.4 | 83.2 | 84.0 | 84.0 |
| AIME24 | 26.7 | 33.0 | 36.7 | 26.7 |
| AIME25 | 23.3 | 30.0 | 30.0 | 26.7 |

**Instructions:**

- Learn from the provided $\{K\}$ numbered explorations and combine their useful reasoning to compute the final answer.
- Produce a single **Execution:** section that carries out the computation and presents the final answer.
- Stop immediately after the final answer. Do not append extra commentary, explanations, or any additional text beyond the required Execution section and the answer.

