# OpenReview forum: "Explore-Execute Chain: Towards an Efficient Structured Reasoning Paradigm"
_ICLR.cc/2026/Conference — ICLR 2026 Conference Withdrawn Submission_

### Official Review · Reviewer_qDxm · 2025-10-28

**Soundness:** 3
**Presentation:** 3
**Contribution:** 2
**Rating:** 4
**Confidence:** 3

**Summary:**

The paper proposes the Explore-Execute Chain (E2C), a reasoning framework that separates planning from execution in large language models. In the exploration phase, the model generates concise, high-level plans; in the execution phase, it deterministically follows the chosen plan. This decoupling improves efficiency, interpretability, and generalization. E2C is trained using a two-stage process combining supervised fine-tuning with a specialized causal data generation method and reinforcement learning to enforce plan adherence and reduce execution variance. Experiments on math and medical benchmarks show that E2C outperforms baselines while using significantly fewer tokens, particularly excelling in test-time scaling and cross-domain adaptation.

**Strengths:**

- The paper is technically rigorous, combining a well-designed causal data generation process, SFT, RL with token-level control to ensure plan adherence and deterministic reasoning.
- The experiments are also comprehensive, spanning mathematical and medical benchmarks, and convincingly demonstrate efficiency gains and cross-domain adaptability.
- The writing is clear and well-structured, with intuitive motivation, formalism, and experimental validation.

**Weaknesses:**

- I think the biggest issue is the novelty and significance. In fact, searching for high-level plans before acting has been extensively explored recently, e.g., PlanSearch [1]. The use of RL does not make significant addition to this test-time computation scaling paradigm.
- The proposed framework is not flexible enough. The authors argue that the monolithic, auto-regressive generation process (CoT) "leads to critical inefficiencies". In fact, CoT uses the intermediate steps as a sketch pad to improve the reasoning performance. The paper completely decouple the high-level planning and low-level execution, which prohibits many established methods built upon CoT, e.g., ReAct, Reflexion, etc.
- A further concern is that the paper introduces a relatively heavy framework to achieve what might be modest empirical gains. The explicit exploration–execution separation, along with the additional training stages and reward mechanisms, seems to add considerable complexity to both training and inference.


[1] Planning In Natural Language Improves LLM Search For Code Generation

**Questions:**

see weaknesses

---

> ### Author Response · Authors · 2025-11-22
> **Response to Reviewer qDxm (Part 1)**
>
> **[W1]**
>
> We appreciate the reviewer’s concern that *“the biggest issue is the novelty and significance”* and that *“searching for high-level plans before acting has been extensively explored (e.g., PlanSearch), and the use of RL does not make significant addition to this test-time computation scaling paradigm.”* We have now explicitly added PlanSearch and related work to the **Related Work** section. While we share the high-level idea of planning before acting, our work is original in how it **formalizes**, **trains**, and **evaluates the efficiency** of this paradigm along three concrete axes:
>
> * **Structured phase definition.** We formalize the **Explore–Execute Chain ($E^2C$)** as a two-stage reasoning framework with clearly separated and complementary phases: a stochastic **exploration phase** that generates informative, high-level plans, and a **deterministic execution phase** that follows the chosen plan. This plan–execute structure is encoded directly into the model (via causal SFT and delimiters), rather than implemented as an external test-time search heuristic.
>
> * **Training-aware implementation.** We design a **two-stage training pipeline** that combines **$E^2C$-SFT** with **$E^2C$-RL**, where each stage optimizes a distinct objective (learning to produce plans vs. learning to follow them). This yields a controlled and reproducible transition from standard autoregressive reasoning to structured two-stage reasoning, rather than relying solely on prompting or search-time tricks.
>
> * **Efficiency and empirical benefits.** Under this framework, $E^2C$ improves stability, **reduces token cost for domain adaptation** (e.g., EF-SFT uses only **3.5%** of standard SFT tokens), and achieves performance comparable to strong test-time scaling baselines, while improving interpretability by explicitly decoupling planning from execution.
>
> **Regarding the role of RL,** we emphasize that our design is **purposeful rather than ornamental**. **First**, we introduce RL with a **specific goal**: to reinforce the structural properties defined by $E^2C$, i.e., making the exploration phase informative and the execution phase close to deterministic, so that the learned policy truly conforms to our two-stage paradigm rather than just benefiting from more computation. **Second**, our RL design is **intentionally simple**—a standard GRPO-style setup with **advantage scaling applied only to exploration tokens**—so that it is **easy to integrate with other RL methods**. Our additional experiments with **DAPO**, where we plug in the same exploration-only advantage scaling with minimal changes and still observe consistent gains, demonstrate that the RL component is both **aligned with our structural objectives** and **readily pluggable into existing RL pipelines**, rather than gratuitous complexity.
>
> **Table 1: Performance comparison of Qwen3 models using DAPO on mathematical reasoning benchmarks.** All results are reported as Pass@1 accuracy (8-sample average). *DAPO (Advantage Scaled)* applies an advantage scaling factor $\lambda=2$ only to exploration tokens; all other settings follow the default DAPO configuration.
>
> | Model                                               |  AIME'24 |  AIME'25 |  MATH500 |  Algebra |  Minerva |   AMC23  | Olympiad |  Avg Acc |
> | :-------------------------------------------------- | :------: | :------: | :------: | :------: | :------: | :------: | :------: | :------: |
> | **Qwen3 4B Series**                                 |          |          |          |          |          |          |          |          |
> | Qwen3-4B + $E^2C$-SFT + DAPO                        |   30.1   |   28.3   |   86.9   |   85.3   |   34.6   |   77.9   |   56.8   |   57.1   |
> | Qwen3-4B + $E^2C$-SFT + **DAPO (Advantage Scaled)** | **40.0** | **31.6** | **88.2** | **87.2** | **35.2** | **80.3** | **60.2** | **60.4** |
> | **Qwen3 8B Series**                                 |          |          |          |          |          |          |          |          |
> | Qwen3-8B + $E^2C$-SFT + DAPO                        |   41.3   |   35.2   |   86.3   |   89.3   |   35.2   |   80.2   |   59.1   |   60.9   |
> | Qwen3-8B + $E^2C$-SFT + **DAPO (Advantage Scaled)** | **42.1** | **37.1** | **90.2** | **91.7** | **38.1** | **82.4** | **62.3** | **63.4** |

---

> > ### Author Response · Authors · 2025-11-23
> > **Response to Reviewer qDxm (Part 2)**
> >
> > **W2: Flexibility with Dynamic Context**
> >
> > We appreciate the reviewer's insight regarding the dynamic nature of reasoning.
> > We respectfully argue that $E^2C$ does not prohibit the dynamic context utilization seen in ReAct or Reflexion; rather, it stabilizes it through a hierarchical approach that supports both intra-execution adaptation and inter-phase re-planning.
> >
> > * **1. The plan-and-execute framework can sucessfully utilize ReAct and Reflexion methods**
> >
> >     * The reviewer concerns that the Plan is rigid, this is a misunderstanding of the role of the Execution Phase. In $E^2C$, the Execution Phase acts as the context-aware engine. In ReAct, the model generates a thought, calls a tool, observes the result, and then generates the next thought.In $E^2C$, this exact loop occurs within the Execution Phase. When the model executes a plan step, it generates the call, receives the new context (the query result), and then generates the subsequent tokens.
> >
> >    * The key difference is that this generation is conditioned jointly on the New Context (the tool output) AND the Global Plan (the strategic intent).Therefore, we do not lose the ability to use new information; we simply ensure that the interpretation of this new information remains aligned with the original goal, preventing the error propagation common in monolithic ReAct.
> >
> >     * Furthermore, if the tool output contradicts the current step, the Execution phase which is trained via RL to maximize final correctness will possess the capability for **Intra-Execution Adaptation**. It can generate tokens to acknowledge the discrepancy and pivot tactically. This effectively embeds the Reflexion loop directly into the execution stream without breaking the high-level strategic structure.
> >
> > * **2. Multi-Round Extensibility**
> >     For particularly long-horizon reasoning tasks where the initial context is insufficient to form a complete plan, $E^2C$ naturally extends to a **Multi-Round Plan-Execute** paradigm.
> >     * We acknowledge that in complex scenarios, the global plan itself may need updating based on execution outcomes. The framework supports this by treating the output of one *Execution Phase* as the context for a subsequent *Exploration Phase*.
> >     * This allows the model to perform **Global Re-planning**: identifying the next stage's goal based on current progress and generating a fresh plan for the remaining steps.
> >     * This structure scales ReAct from a step-by-step loop to a phase-by-phase cycle (**Plan $\to$ Execute $\to$ Observe $\to$ Re-Plan**), offering significantly better stability for long-context reasoning by compartmentalizing the thinking space.

---

> > > ### Author Response · Authors · 2025-11-23
> > > **Response to Reviewer qDxm (Part 3)**
> > >
> > > **[W3]**
> > >
> > > We fully understand the concern that our framework involves a multi-stage process. However, we respectfully argue that this structural complexity is not mere overhead; it is the specific mechanism that unlocks significant reductions in computational and data costs during both inference and adaptation.
> > >
> > > * **1. Minimal Training Cost:**
> > >     We specifically investigated training-free approaches and found them unstable for enforcing strict plan adherence. To address this, we designed our training stages to be as lightweight as possible.
> > >     * **E$^2$C-SFT:** This stage requires only a minimal dataset and completes rapidly.
> > >     * **E$^2$C-RL:** As noted in our response to the training complexity concern, our RL stage converges using approximately 80% of the rollouts required by standard GRPO.
> > >     Thus, we achieve the necessary stability for structured reasoning with a training budget comparable to, or even more efficient than, standard baselines.
> > >
> > > * **2. Structure-Driven Efficiency:**
> > >     We emphasize that maximizing absolute performance metrics was not our sole objective; rather, we prioritize the systematic benefits brought by **structured reasoning**. The explicit separation of exploration and execution unlocks efficiencies that monolithic models cannot achieve:
> > >     * **Inference Efficiency:** As shown in **Figure 1** and **Table 3**, our Test-Time Scaling strategy achieves 58.1% accuracy using **less than 10%** of the decoding tokens required by comparable methods like Forest-of-Thought (12.1k vs. 128.8k tokens).
> > >     * **Adaptation Efficiency:** This structure drastically lowers data requirements for transfer. Our **EF-SFT** method (Table 2) requires only **10M tokens** (3.5% of the standard SFT budget) to outperform the standard SFT baseline by **14.5%** on Llama3.1-8B.
> > >     * **Robustness:** Even though absolute gain is not the primary focus, the structure remains robust. As shown in **Table 1** provided in the **[W1]** rebuttal, adding E$^2$C to the strong **DAPO** baseline improves performance by **4.5%** on Qwen3-4B, proving that the structure contributes consistent value.
> > >
> > > * **3. Interactive Inference:**
> > >     The structured *Plan-First* output offers unique advantages for extensibility and user interaction that monolithic models lack. The model can output the Plan for user verification before committing to the expensive Execution phase. If the generated Plan is visibly incorrect, the inference process can be terminated early, saving significant computational resources. Therefore, the structural overhead in training translates into a highly flexible, controllable, and efficient inference engine in deployment.

---

> ### Author Response · Authors · 2025-11-28
> **Further discussion with Reviewer qDxm**
>
> Dear Reviewer qDxm,
>
> We sincerely thank you for your careful feedback. We have addressed your concerns and summarize our responses as follows:
>
> 1. Novelty and significance: While prior work explores planning before acting (e.g., PlanSearch), $E^2C$ is original in formalizing, training, and evaluating a structured two-stage reasoning framework. Exploration generates informative high-level plans; execution deterministically follows them. RL is purposefully applied to reinforce plan quality and adherence, with advantage scaling on exploration tokens, and is plug-and-play with DAPO, empirically improving performance (e.g., Table 1).
>
> 2. Flexibility and dynamic reasoning: $E^2C$ supports intra-execution adaptation and ReAct/Reflexion-style loops within the Execution Phase. Execution conditions on both the global plan and new observations, allowing tactical pivots. Multi-round Plan–Execute enables global re-planning, extending ReAct from step-by-step to phase-by-phase reasoning, preserving stability while using dynamic context.
>
> 3. Framework complexity vs. efficiency: The two-stage design reduces computational and data costs while maintaining robustness. EF-SFT uses only 3.5% of standard SFT tokens to outperform baselines; Test-Time Scaling achieves high accuracy with <10% decoding tokens. Plan verification before execution enables controlled, efficient inference. Structured training overhead translates into a flexible, interpretable, and efficient reasoning engine.
>
> As the author–reviewer discussion phase is drawing to a close, we would like to confirm whether our responses adequately address your concerns. If you have any further questions, please feel free to let us know.
>
> Thank you again for your thoughtful review and time.
>
> Best regards,
> Authors

---

### Official Review · Reviewer_AJeZ · 2025-10-31

**Soundness:** 2
**Presentation:** 3
**Contribution:** 2
**Rating:** 4
**Confidence:** 4

**Summary:**

The paper proposes the Explore–Execute Chain (E2C) framework, which separates reasoning into an exploratory phase for generating high-level plans and an execution phase for detailed reasoning. Using a two-stage training process with supervised fine-tuning and reinforcement learning, E2C aims to improve efficiency and interpretability. Experiments on math and medical reasoning benchmarks show improvements in accuracy with reduced computational cost and efficient domain adaptation through the Exploration-Focused SFT method.

**Strengths:**

- The paper is clearly presented and well organized.

- The experiments are comprehensive, and the authors provide implementation details and intermediate results.

- The advantage in improving efficiency is clear.

**Weaknesses:**

- The proposed plan–then–execute paradigm raises concerns regarding its effectiveness. In our experience, prompting a model to first produce a plan and then execute it often yields worse results than directly applying CoT prompting. This limitation stems from the fact that the model must generate a plan based solely on the query, without access to any intermediate reasoning results, which may hinder its ability to form accurate or adaptive strategies.

- According to the official report [1] (Table 18), simply extending the CoT length allows Qwen3-8B to achieve 76.0 on AIME’24 and 67.3 on AIME’25, substantially higher than the 40.6 and 33.8 reported by the proposed method despite its added complexity, including additional training stages, hyperparameter tuning, and structural modifications. This comparison raises questions about the significance of the proposed approach relative to simpler alternatives.

- The proposed E2C-RL algorithm is quite similar to Hierarchy-Aware Credit Assignment [2], differing mainly in placing all planning tokens at the beginning, which limits its novelty.

- The domain adaptation experiment setup seems unfair. The standard SFT baseline is fine-tuned solely on medical data, whereas the proposed method uses both math and medical data. A fairer comparison is to involve math data in standard SFT baseline as well.

- Code and data are not provided for review.


## Reference:
[1] Yang, An, et al. "Qwen3 technical report." arXiv preprint arXiv:2505.09388 (2025).

[2] Wang, Haozhe, et al. "Emergent hierarchical reasoning in llms through reinforcement learning." arXiv preprint arXiv:2509.03646 (2025).

**Questions:**

- Could the authors include results for standard SFT baselines trained on both math and medical data to enable a fairer comparison with the proposed EF-SFT method?

- Could the authors clarify the notations used in Section 3.2.2? For example, what do the indices $i$ and $t$ represent, how are $T_{\text{exp}}$ and $T_{\text{exe}}$ determined, and what does $H$ denote in Equation (7)?

- Could the authors provide the code and data for review?

---

> ### Author Response · Authors · 2025-11-22
> **Response to Reviewer AJeZ (Part 1)**
>
> **[W1, W2]**
>
> We appreciate the reviewers’ concerns regarding both the **effectiveness** of the generated plans and their **ability to be dynamically adjusted** during execution. We address these points from two complementary perspectives:
>
> * **Plan quality.** On 370 IMO problems, we generate 5 plans per problem, yielding a total of 1,850 plans. Only 90 of these plans are incorrect when compared with the reference solutions, indicating that the vast majority of our plans are accurate and reliable.
>
> * **Plan sensitivity.**
>   We further validate the importance of plan quality by **fixing the executor to the GRPO-trained Qwen3-8B+$E^2C$-(SFT+RL) model** and only swapping the planning module. Specifically, for each test instance, we compare execution using plans generated by the **Qwen3-8B+$E^2C$-(SFT+RL)** model and by its **Qwen3-8B+$E^2C$-SFT** variant, while keeping all other settings unchanged. When replacing the GRPO-trained plans with SFT-only plans, execution accuracy drops by about **7 percentage points on AIME'24**, about **4 percentage points on AIME'25**, and about **4 percentage points on MATH500**. These results indicate that execution performance is highly sensitive to plan quality and that RL training substantially improves the usefulness of the generated plans.
>
>   | Plan source                         |  AIME'24 |  AIME'25 |  MATH500 |
>   | ----------------------------------- | :------: | :------: | :------: |
>   | Qwen3-8B+$E^2C$-SFT as planner      |   33.8   |   29.8   |   83.4   |
>   | Qwen3-8B+$E^2C$-(SFT+RL) as planner | **40.6** | **33.8** | **87.7** |
>
> These findings collectively demonstrate that $E^2C$ produces **high-quality, execution-critical plans**. At the same time, the framework is designed to support **dynamic plan adjustment** when needed:
>
> * **Spontaneous reflection.** During RL training, the reward design does not penalize the model for revising or extending its reasoning after following the planned steps, as long as it respects the exploration tokens. This allows the model to engage in spontaneous self-reflection and correction while still enforcing adherence to the initial plan structure.
>
> * **External plan editing.** The framework also supports explicit plan refinement by adding dedicated steps in the exploration phase (e.g., “*n+1. Reflect on and revise the plan*”). This provides a controlled mechanism to update or improve the plan before execution, while preserving the clear separation between planning and execution.
>
> Together, these mechanisms show that $E^2C$ not only generates reliable plans that strongly influence execution quality, but also allows for flexible and principled dynamic adjustment when the task demands it.
>
> **[W3]**
>
> We recognize that prior work such as Hierarchy-Aware Credit Assignment [2] is meaningful, as it demonstrates that careful credit assignment over planning tokens can improve performance. Our $E^2C$-RL algorithm differs mainly in its integration with a fully structured two-stage reasoning paradigm: by explicitly placing all planning (exploration) tokens before execution tokens and separating them with a special delimiter `</EXPLORATION>`, we facilitate straightforward advantage scaling. This design allows easy integration with existing RL frameworks while reinforcing the intended behavior of the exploration and execution phases, which is a structural benefit rather than a mere modification of GRPO or similar methods.
>
> To further validate this, we conducted additional experiments integrating $E^2C$ with DAPO. We applied an advantage scaling factor $\lambda=2$ to exploration tokens while keeping all other DAPO components unchanged. The results, shown in Table 1, confirm the effectiveness of this approach and the natural plug-and-play property enabled by our structured reasoning design.
>
> **Table 1: Performance comparison of Qwen3 models using DAPO on mathematical reasoning benchmarks.** All results are reported as Pass@1 accuracy (8-sample average). *DAPO (Advantage Scaled)* indicates that an advantage scaling factor $\lambda=2$ is applied only to exploration tokens, while all other settings follow the default DAPO configuration.
>
> | Model | AIME'24 | AIME'25 | MATH500 | Algebra | Minerva | AMC23 | Olympiad | Avg Acc |
> | :--- | :---: | :---: | :---: | :---: | :---: | :---: | :---: | :---: |
> | **Qwen3 4B Series** | | | | | | | | |
> | Qwen3-4B + $E^2C$-SFT + DAPO | 30.1 | 28.3 | 86.9 | 85.3 | 34.6 | 77.9 | 56.8 | 57.1 |
> | Qwen3-4B + $E^2C$-SFT + DAPO (Advantage Scaled) | 40.0 | 31.6 | 88.2 | 87.2 | 35.2 | 80.3 | 60.2 | 60.4 |
> | **Qwen3 8B Series** | | | | | | | | |
> | Qwen3-8B + $E^2C$-SFT + DAPO | 41.3 | 35.2 | 86.3 | 89.3 | 35.2 | 80.2 | 59.1 | 60.9 |
> | Qwen3-8B + $E^2C$-SFT + DAPO (Advantage Scaled) | 42.1 | 37.1 | 90.2 | 91.7 | 38.1 | 82.4 | 62.3 | 63.4 |

---

> > ### Author Response · Authors · 2025-11-22
> > **Response to Reviewer AJeZ (Part 2)**
> >
> > **[W4]**
> >
> > We carefully address the reviewer’s concern on the adherence between exploration and execution, which is a central issue in our two-stage paradigm, through the following stage-specific designs:
> >
> > * **During the SFT stage**, we enforce a strict causal dependency from exploration to execution by adopting a two-step data generation pipeline, where the model is first prompted to generate an exploration and then required to produce the execution *conditioned solely on this exploration*, instead of extracting a plan from an already sampled execution, so as to avoid shortcut learning; the effectiveness of this SFT data construction strategy is empirically validated in Table 4 (Part A).
> > * **During the RL stage**, we explicitly quantify and encourage plan-execution adherence by imposing a deterministic formatting constraint in our SFT data, where the exploration plan must be organized as an enumerated list (e.g., “1.xxx, 2.xxx, …”) and the execution is required to restate each corresponding plan step, which allows us to compute the adherence score between exploration and execution using regular expressions, and use this score as an *instruction reward* in **$E^2C$-RL** to guide the model to strictly follow its generated exploration; the detailed procedure is described in Appendix A.2.2. We thank the reviewer for highlighting this critical issue.
> >
> >
> > **[Q1]**
> >
> > We acknowledge the reviewer's careful consideration of domain adaptation. Following this suggestion, we conducted additional experiments comparing joint fine-tuning (math + medical) and medical-only SFT. As shown in **Table 2**, for both Qwen3-8B and Llama3.1-8B, joint fine-tuning with 10% math and 90% medical data does *not* improve performance over medical-only SFT, and instead introduces unnecessary complexity and slight performance drops. Therefore, our design uses medical-only SFT for domain adaptation while reserving mathematical data for structured reasoning training in $E^2C$. This choice is empirically justified and methodologically clean.
> >
> > **Table 2: Performance comparison of SFT models under different domain adaptation strategies.** Rows labeled with *(10% Math + 90% Medical)* correspond to the joint tuning baselines requested. Columns AN, CK, CB, CM, MG, and PM are validation subsets of the MMLU benchmark.
> >
> > | Model | MedQA | MedMCQA | AN | CK | CB | CM | MG | PM | #Med-Tokens | Avg |
> > | :--- | :---: | :---: | :---: | :---: | :---: | :---: | :---: | :---: | :---: | :---: |
> > | ***SFT Models (using medical data)*** | | | | | | | | | | |
> > | Qwen3-8B + standard SFT (10% Math + 90% Medical) | 53.2 | 51.3 | 63.8 | 75.8 | 83.1 | 68.2 | 73.3 | 75.3 | 286M | 68.0 |
> > | Qwen3-8B + standard SFT (100% Medical) | 58.2 | 52.3 | 68.8 | 80.8 | 89.0 | 73.7 | 83.3 | 79.0 | 286M | 73.1 |
> > | Llama3.1-8B + standard SFT (10% Math + 90% Medical) | 38.0 | 34.2 | 46.2 | 51.2 | 57.9 | 41.3 | 35.1 | 57.5 | 286M | 45.2 |
> > | Llama3.1-8B + standard SFT (100% Medical) | 42.0 | 36.8 | 45.9 | 55.4 | 61.8 | 43.2 | 38.1 | 56.9 | 286M | 47.5 |
> > | ***EF-SFT Models*** | | | | | | | | | | |
> > | Qwen3-8B + $E^2C$-SFT + EF-SFT | 65.8 | 58.2 | 72.3 | 83.8 | 89.2 | 79.7 | 87.6 | 86.2 | 10M | 77.1 |
> > | Llama3.1-8B + $E^2C$-SFT + EF-SFT | 60.3 | 53.2 | 61.8 | 69.8 | 75.9 | 64.9 | 82.0 | 72.5 | 10M | 67.5 |
> >
> > **[Q2]**
> >
> > We acknowledge the reviewer's note regarding the imprecise expression and have corrected the relevant parts in the PDF (highlighted in green). To clarify the issue raised: in the formula, $H$ denotes entropy; the plan tokens and execution tokens are separated by the special token `</EXPLORATION>`. The subscripts $i$ and $t$ indicate the $i$-th rollout in the batch and the $t$-th token within that rollout, respectively.
> >
> > **[W5,Q3]**
> >
> > We appreciate the reviewer’s attention. Our code and data are publicly available at [https://anonymous.4open.science/r/Explore-Execute-Chain-5C1F/](https://anonymous.4open.science/r/Explore-Execute-Chain-5C1F/).

---

> ### Author Response · Authors · 2025-11-28
> **Further discussion with Reviewer AJeZ**
>
> Dear Reviewer AJeZ,
>
> We sincerely thank you for your careful feedback. We have addressed your concerns and summarize our responses as follows:
>
> 1. Regarding plan–then–execute effectiveness and dynamic adjustment: on 370 IMO problems, we generated 1,850 plans with only 90 errors. Swapping RL-trained planners with SFT-only planners drops execution accuracy by 7 / 4 / 4 percentage points on AIME’24 / AIME’25 / MATH500, showing plan quality strongly affects execution. Dynamic adjustment is supported via spontaneous reflection during RL and explicit plan revision in the exploration phase.
>
> 2. Regarding novelty vs. Hierarchy-Aware Credit Assignment: $E^2C$-RL differs through its fully structured two-stage design, placing all planning tokens first and separating them with a delimiter. This allows advantage scaling on exploration tokens and enables plug-and-play integration with DAPO, which we empirically validate.
>
> 3. Regarding plan–execution adherence: during SFT, execution is strictly conditioned on generated plans; during RL, structured enumerated plan formats and regex-based adherence rewards explicitly encourage execution to follow exploration plans.
>
> 4. Regarding domain adaptation fairness: joint fine-tuning with 10% math and 90% medical data does not improve over medical-only SFT and can slightly reduce performance. Therefore, we use medical-only SFT for domain adaptation while reserving mathematical data for structured reasoning in $E^2C$.
>
> 5. Regarding notation and code/data availability: in Section 3.2.2, $i$ is the rollout index, $t$ is the token index, $H$ denotes entropy, and plan/execution tokens are separated by `</EXPLORATION>`. Code and data are publicly available at https://anonymous.4open.science/r/Explore-Execute-Chain-5C1F/
>
>
> As the author–reviewer discussion phase is drawing to a close, we would like to confirm whether our responses adequately address your concerns. If you have any further questions, please feel free to let us know.
>
> Thank you again for your thoughtful review and time.
>
> Best regards,
> Authors

---

### Official Review · Reviewer_A4xv · 2025-11-01

**Soundness:** 2
**Presentation:** 3
**Contribution:** 2
**Rating:** 4
**Confidence:** 4

**Summary:**

The paper proposes the Explore–Execute Chain (E2C) framework, which decomposes reasoning into two phases: an exploratory stage that generates high-level plans and an execution stage that deterministically follows these plans. The method is trained through a two-stage process combining supervised fine-tuning and reinforcement learning. Experiments on mathematical and medical reasoning benchmarks show that E2C improves efficiency and maintains competitive accuracy compared to existing structured reasoning approaches.

**Strengths:**

The paper presents a clear and organized framework that separates reasoning into exploration and execution phases. The experiments are reasonably comprehensive and the writing is clear. The methodology is well presented.

**Weaknesses:**

The core idea of strictly separating planning and execution lacks sufficient justification, as it prevents the model from adapting its plan mid-way, which is often crucial for complex reasoning tasks. In addition, generating effective plans without preliminary computation remains challenging. The proposed GRPO-based training closely resembles existing work [1], which limits the novelty of the methodology.

[1] Wang, Haozhe, et al. "Emergent hierarchical reasoning in llms through reinforcement learning." arXiv preprint arXiv:2509.03646 (2025).

**Questions:**

• The proposed framework enforces a strict separation between planning and execution, which prevents the model from adjusting its plan mid-way. This could potentially degrade performance on complex tasks where adaptive planning is necessary. How do the authors address or mitigate this limitation?

• How effective are the generated plans in practice? For challenging mathematical problems, forming a correct plan often requires partial calculations or intermediate reasoning. How does the model ensure that plans remain useful without preliminary computation?

• Since the plan omits details and merely serves as part of the context, there is no guarantee that the execution phase will fully adhere to it. How do the authors ensure consistency and faithful plan adherence during execution?

• Could the authors clarify what are the exploration and exploitation tokens in their RL formulation? Are these sets equivalent to the plan tokens $\pi$ and execution tokens $e$, respectively?

• How robust is the clustering-based selection method at test time? Given that small semantic differences may not significantly affect embedding similarity, how do the authors ensure that clustering reliably distinguishes meaningfully different plans?

---

> ### Author Response · Authors · 2025-11-22
> **Response to Reviewer A4xv (Part 1)**
>
> **[Q1]**
>
> We appreciate this insightful point. Our framework supports dynamic plan adjustment through multiple mechanisms while maintaining the benefits of structured reasoning:
>
> *  **Spontaneous reflection in RL:** During RL training, after executing the initial plan, the model naturally engages in reflection and self-correction. This shows that structured execution does not preclude adaptive planning.
> *  **Explicit plan editing:** The exploration phase allows direct plan refinement by inserting steps such as *n+1. Reflect and revise the plan*. This provides controlled flexibility without compromising the two-phase paradigm.
>
> These mechanisms ensure that the framework preserves structured reasoning while accommodating the adaptive planning required for complex tasks.
>
> **[Q2]**
> We appreciate the reviewer’s question. We add two experiments to demonstrate the high quality and effectiveness of the generated plans from two perspectives:
>
> * **Plan quality.** We conduct an additional experiment on the IMO dataset, where we sample five trajectories (plans) for each problem and compare them with the ground-truth solutions to check for planning errors. The evaluation shows that the vast majority of plans are correct: only 90 out of 1,850 sampled plans contain errors. This confirms that our planning approach reliably produces high-quality plans even on challenging IMO-level problems (see the attached file `imo.json` for the raw data).
>
> * **Plan sensitivity.** We further validate the importance of plan quality by **fixing the executor to the GRPO-trained Qwen3-8B+$E^2C$-(SFT+RL) model** and only swapping the planning module. Specifically, for each test instance, we compare execution using plans generated by the **Qwen3-8B+$E^2C$-(SFT+RL) model** and by its **Qwen3-8B+$E^2C$-SFT** variant, while keeping all other settings unchanged. When replacing the GRPO-trained plans with SFT-only plans, execution accuracy drops by about **7 percentage points on AIME'24**, about **4 percentage points on AIME'25**, and about **4 percentage points on MATH500**. These results indicate that actual performance is highly sensitive to plan quality.
>
> | Plan source                              | AIME'24 | AIME'25 | MATH500 |
> |------------------------------------------|:-------:|:-------:|:-------:|
> | Qwen3-8B+$E^2C$-SFT as planner              |  33.8   |  29.8   |  83.4   |
> | Qwen3-8B+$E^2C$-(SFT+RL) as planner         | **40.6**| **33.8**| **87.7**|
>
>
> These findings collectively demonstrate that $E^2C$ generates high-quality, reliable plans that make a substantial and measurable contribution to successful execution.
>
> **[Q3]**
>
> We carefully address the reviewer’s concern on the adherence between exploration and execution, through the following stage-specific designs.
>
> * **During the SFT stage**, we enforce a strict causal dependency from exploration to execution by adopting a two-step data generation pipeline, where the model is first prompted to generate an exploration and then required to produce the execution *conditioned solely on this exploration*, instead of extracting a plan from an already sampled execution, so as to avoid shortcut learning; the effectiveness of this SFT data construction strategy is empirically validated in Table 4 (Part A).
> * **During the RL stage**, we explicitly quantify and encourage plan-execution adherence by imposing a deterministic formatting constraint in our SFT data, where the exploration plan must be organized as an enumerated list (e.g., “1.xxx, 2.xxx, …”) and the execution is required to restate each corresponding plan step, which allows us to compute the adherence score between exploration and execution using regular expressions, and use this score as an *instruction reward* in **$E^2C$-RL** to guide the model to strictly follow its generated exploration; the detailed procedure is described in Appendix A.2.2.

---

> > ### Author Response · Authors · 2025-11-22
> > **Response to Reviewer  A4xv (Part 2)**
> >
> > **[Q4]**
> >
> > We thank the reviewer for the clarification. In our RL formulation, the *exploration tokens* correspond exactly to the plan tokens $\pi$, and the *exploitation tokens* correspond to the execution tokens $e$. In implementation, we explicitly separate these two segments with a special delimiter token `</EXPLORATION>`, which allows us to cleanly distinguish exploration from execution during training (e.g., masking, log-prob computation) and to assign rewards to the appropriate part of the trajectory.
> >
> > **[Q5]**
> >
> > We thank the reviewer for this important question regarding the robustness of our clustering procedure. We address this concern from two complementary angles:
> >
> > 1. **High-level plans and semantically aware encoders.**
> >    Our framework is designed to produce high-level exploration plans rather than low-level calculation steps. After numeric details and intermediate computations are stripped away, the remaining text primarily encodes strategic choices (e.g., “apply conservation of energy” vs. “use kinematic equations”), which naturally leads to distinct semantic embeddings. We further employ the all-mpnet-base-v2 encoder, which is optimized to map semantically similar but syntactically different sentences to nearby representations, thereby reducing sensitivity to superficial phrasing changes.
> >
> > 2. **Empirical evidence via performance gap.**
> >    The robustness of the clustering is also supported by the clear performance gap between our *Semantic Cluster* selection and the *Random Plan* baseline. If the embedding-based clustering were dominated by noise or only minor variations, its behavior would approach random selection and the performance advantage would largely disappear. The observed improvement therefore provides practical evidence that the clustering captures meaningfully different strategies rather than incidental wording differences.

---

> ### Author Response · Authors · 2025-11-28
> **Further discussion with Reviewer A4xv**
>
> Dear Reviewer A4xv,
>
> We sincerely thank you for your thoughtful feedback. We have carefully addressed your concerns and summarize our responses as follows:
>
> 1. Although our framework enforces a plan–execute separation, it still supports adaptive planning through spontaneous reflection during RL and explicit plan editing in the exploration phase, preserving structure while enabling flexibility.
>
> 2. We provide new evidence on plan quality and usefulness: only 90 errors out of 1,850 sampled IMO plans(high-quality); Replacing RL-trained planners with SFT-only planners leads to clear performance drops (e.g., −7% on AIME’24), demonstrating that execution performance is highly sensitive to plan quality.
>
> 3. To ensure plan adherence, we enforce causal conditioning in SFT and introduce a plan–execution alignment reward in RL, based on structured plan formatting and regex matching.
>
> 4. We clarify that exploration tokens correspond to plan tokens π and exploitation tokens to execution tokens e, explicitly separated by a delimiter during training.
>
> 5. For clustering, we use a semantically robust encoder and observe clear gains over random plan selection, indicating that clustering reliably captures meaningful strategic differences.
>
> As the author–reviewer discussion phase is drawing to a close, we would appreciate your thoughts on whether these clarifications address your concerns.
>
> Thank you again for your time and valuable feedback.
>
> Best regards,
> Authors

---

### Official Review · Reviewer_Zd2Z · 2025-11-01

**Soundness:** 3
**Presentation:** 2
**Contribution:** 2
**Rating:** 4
**Confidence:** 3

**Summary:**

This paper introduces Explore–Execute Chain (E2C), a reasoning framework that separates planning (exploration) from execution in large language models. With a two-stage SFT + RL training method, E2C achieves higher efficiency and interpretability, reaching 58.1% accuracy on AIME’2024 with under 10% of the tokens and enabling data-efficient domain adaptation across tasks.

**Strengths:**

This paper introduces the Explore–Execute Chain (E2C), a novel framework that separates high-level planning from detailed execution to enhance reasoning efficiency and interpretability. Through a carefully designed two-stage SFT+RL training pipeline and a data-efficient adaptation method (EF-SFT), E2C achieves state-of-the-art results with far fewer tokens, demonstrating strong generalization, scalability, and clear methodological innovation.

**Weaknesses:**

1. The idea of separating reasoning into “plan” and “execute” phases is not very novel, as similar frameworks have been explored in prior works such as Planning in Natural Language Improves LLM Search for Code Generation. The main contribution lies in applying RL to this paradigm, which feels incremental.

2. The performance improvements over strong baselines (e.g., GRPO) are relatively small—around 1–2% in Tables 1 and 2—making it hard to clearly demonstrate the benefits of the proposed approach.

3. The baseline comparison is limited. Important RL-based methods such as DAPO and RLOO are missing, which weakens the empirical validation.

4. While the paper claims efficiency, the RL training pipeline appears complex and resource-intensive. The actual training cost versus GRPO or standard SFT is not quantified.

**Questions:**

see weaknesses.

---

> ### Author Response · Authors · 2025-11-22
> **Response to Reviewer Zd2Z**
>
> **[W1]**
>
> We thank the reviewer for pointing out this prior work on plan–execute decomposition (e.g., *Planning in Natural Language Improves LLM Search for Code Generation*), and we have now explicitly added it to the **Related Work** section. While we share the high-level idea of separating planning from execution, our work is original in how it formalizes, trains, and empirically demonstrates the efficiency of this paradigm:
>
> * **Structured phase definition.** We formalize the **Explore–Execute Chain ($E^2C$)** as a two-stage reasoning framework with clearly separated and complementary phases. The *exploration phase* stochastically generates informative, high-level plans, whereas the *execution phase* deterministically carries out the chosen plan. This strict separation enforces both effective exploration and reliable execution.
>
> * **Training-aware implementation.** We design a two-stage training pipeline that combines **Supervised Fine-Tuning ($E^2C$-SFT)** with subsequent **Reinforcement Learning ($E^2C$-RL)**. Each stage explicitly optimizes its designated objective, enabling a controlled and reproducible transition from standard autoregressive reasoning to structured two-stage reasoning.
>
> * **Efficiency and empirical benefits.** The $E^2C$ paradigm improves robustness and stability, reduces token cost for domain adaptation (e.g., EF-SFT uses only 3.5% of standard SFT tokens), and achieves performance comparable to strong test-time scaling baselines, while improving interpretability by decoupling planning from execution.
>
> Taken together, these elements constitute a novel contribution: we enforce a training-aware, two-stage structured paradigm and systematically explore its efficiency and performance benefits, providing the first fully training-aware and interpretable realization of plan–execute reasoning in large language models.
>
>
> **[W2,W3]**
>
> We recognize the reviewer's concern regarding the relatively modest performance gains obtained through RL. Our *advantage scaling* technique is inherently modular and can be readily integrated with existing RL frameworks. To validate this, we applied it on top of GRPO. We also consider the reviewer's suggestion to include DAPO as a baseline well-justified, and accordingly conducted additional experiments using DAPO. In these experiments, the results are summarized in Table 1. For example, GRPO gains approximately 3.8% on 8B and 3.2% on 4B, while DAPO achieves approximately 3.3% improvement on 8B and 4.5% on 4B when advantage scaling is applied. These results confirm that our method can be easily incorporated into standard RL approaches while consistently improving performance through structured allocation of exploration rewards.
>
> **Table 1: Performance comparison of Qwen3 models using DAPO on mathematical reasoning benchmarks.** All results are reported as Pass@1 accuracy (8-sample average). The row labeled *DAPO (Advantage Scaled)* indicates that an advantage scaling factor $\lambda=2$ was applied to exploration tokens, while all other settings follow the default DAPO configuration.
>
> | Model | AIME'24 | AIME'25 | MATH500 | Algebra | Minerva | AMC23 | Olympiad | Avg Acc |
> | :--- | :---: | :---: | :---: | :---: | :---: | :---: | :---: | :---: |
> | **Qwen3 4B Series** | | | | | | | | |
> | Qwen3-4B + $E^2C$-SFT + DAPO | 30.1 | 28.3 | 86.9 | 85.3 | 34.6 | 77.9 | 56.8 | 57.1 |
> | Qwen3-4B + $E^2C$-SFT + DAPO (Advantage Scaled) | 40.0 | 31.6 | 88.2 | 87.2 | 35.2 | 80.3 | 60.2 | 60.4 |
> | **Qwen3 8B Series** | | | | | | | | |
> | Qwen3-8B + $E^2C$-SFT + DAPO | 41.3 | 35.2 | 86.3 | 89.3 | 35.2 | 80.2 | 59.1 | 60.9 |
> | Qwen3-8B + $E^2C$-SFT + DAPO (Advantage Scaled) | 42.1 | 37.1 | 90.2 | 91.7 | 38.1 | 82.4 | 62.3 | 63.4 |
>
> The **main benefit** of our RL design is to enforce a *structured reasoning process*, where the exploration phase becomes *informative* and the execution phase *deterministic*. The improvements beyond the base model and standard RL baselines are a *by-product* rather than the design target. Our core objective is to realize the two-phase paradigm we define and to demonstrate its efficiency. In this setting, RL serves to support and stabilize the paradigm, while gains in accuracy and efficiency arise as consequences.
>
> **[W4]**
>
> We note the reviewer’s concern about the complexity and resource demands of our RL pipeline. Compared with standard GRPO, $E^2C$-RL introduces **no extra algorithmic complexity**: in our experiments we use only 80% of the GRPO rollouts to reach the reported performance, and the advantage-scaling step adds negligible overhead, ensuring a fair and efficient comparison.
>
> For EF-SFT, we use only **3.5%** of the tokens of standard SFT for domain adaptation. In practice, EF-SFT on 50k medical datas for Qwen3-8B finishes in about one hour on 8×H800 GPUs, and RL training completes in roughly two days on the same hardware. Thus, $E^2C$ delivers structured reasoning benefits with **modest computational cost** and strong practical efficiency.

---

> ### Author Response · Authors · 2025-11-28
> **Further discussion with Reviewer Zd2Z**
>
> Dear Reviewer Zd2Z,
>
> We sincerely thank you for your detailed and constructive feedback. We have carefully addressed your concerns and provide the following summary of our responses:
>
> 1. We clarify that while the plan–execute paradigm has appeared in prior work, our contribution lies in a training-aware and formalized Explore–Execute Chain （E$^2$C） with a strictly separated stochastic exploration phase and deterministic execution phase, along with an end-to-end SFT + RL training pipeline. We have now explicitly added the cited work to the Related Work section for completeness.
>
> 2. Although the absolute gains over strong baselines appear modest, we emphasize that our objective is to enforce structured reasoning and improve stability, rather than to maximize raw accuracy alone. We further demonstrate that our advantage scaling mechanism consistently improves both GRPO and DAPO (e.g., +3.8% on 8B and +4.5% on 4B), confirming its generality beyond a single RL framework.
>
> 3. To strengthen the baseline comparisons, we have added additional experiments with DAPO, showing consistent improvements under our advantage scaling mechanism. These results verify that our method is compatible with and beneficial to existing RL paradigms, addressing the concern regarding limited empirical validation.
>
> 4. Regarding training efficiency, we clarify that E$^2$C-RL introduces no extra algorithmic complexity compared to GRPO. In practice, we use only 80% of GRPO rollouts to reach comparable or better performance. In addition, the EF-SFT stage requires only 3.5% of standard SFT tokens, significantly reducing domain adaptation cost.
>
> 5. We restate that the primary contribution of our work is not incremental accuracy gains alone, but the first training-aware realization of a structured two-stage reasoning paradigm for LLMs, offering improved interpretability, stability, and efficiency, with performance improvements as a natural consequence.
>
> As the author–reviewer discussion phase is drawing to a close, we would like to confirm whether our responses adequately address your concerns. If you have any further questions, please feel free to let us know.
>
> Thank you again for your thoughtful review and time.
>
> Best regards,
> Authors

---

### Note · Authors · 2025-12-01

I have read and agree with the venue's withdrawal policy on behalf of myself and my co-authors.